

# Solid hydrometeor classification and riming degree estimation from pictures collected with a Multi-Angle Snowflake Camera

Christophe Praz[1], Yves-Alain Roulet[2], and Alexis Berne[1]

[1]Environmental Remote Sensing Laboratory (LTE), École polytechnique Fédérale de Lausanne (EPFL), Lausanne, Switzerland
[2]Federal Office of Meteorology and Climatology MeteoSwiss, Payerne, Switzerland

*Correspondence to:* A. Berne (alexis.berne@epfl.ch)

**Abstract.** A new method to automatically classify solid hydrometeors based on a Multi-Angle Snowflake Camera (MASC) images is presented. For each individual image, the method relies on the calculation of a set of geometric and texture-based descriptors to simultaneously identify the hydrometeor type (among six predefined classes), estimate the degree of riming and detect melting snow. The classification tasks are achieved by means of a regularized multinomial logistic regression (MLR) model trained over more than 3000 MASC images manually labeled by visual inspection. In a second step, the probabilistic information provided by the MLR is weighed on the three stereoscopic views of the MASC in order to assign a unique label to each hydrometeor. The accuracy and robustness of the proposed algorithm is evaluated on data collected in the Swiss Alps and in Antarctica. The algorithm achieves high performance, with a hydrometeor type classification accuracy and Heidke skill score of 95% and 0.93, respectively. The degree of riming is evaluated by introducing a riming index ranging between zero (no riming) and one (graupel), and characterized by a probable error of 5.3%. A validation study is conducted through a comparison with an existing classification method based on two-dimensional video disdrometer (2DVD) data and shows that the two methods are consistent.

## 1 Introduction

Falling hydrometeors can be seen as the signature of the microphysical processes controlling the formation and evolution of precipitation in the atmosphere, and as such it is of primary importance to collect detailed information about them. In particular, the quantitative estimation of precipitation rate using remote sensing techniques or numerical weather prediction (NWP) Models requires knowledge about the microstructure of rain/snow. In the case of snowfall, an accurate modeling of the scattering properties of snowflakes is necessary for the interpretation of radar retrievals and the correct estimation of snowfall rate (e.g., Matrosov, 2007). These properties are strongly influenced by the mass, size and morphology of the particles, as reported by several scattering simulation studies (e.g., Hong, 2007; Petty and Huang, 2010; Johnson et al., 2012). Precipitation rates from NWP models are also strongly affected by the type of hydrometeors (Garvert et al., 2005) considered as well as by the parameterizations of the mass-size and velocity-size relationships (Woods et al., 2007). These parameterizations generally rely on in-situ measurements made in the past under potentially different environmental conditions, and with limited accuracy compared to what can be achieved nowadays. Both from a weather radar and NWP model point of view, it is therefore essential



to document the microstructural properties of individual falling snowflakes in order to better characterize the microphysics of snowfall as well as to improve its quantitative estimation.

Hydrometeor classification methods aim to categorize precipitation in qualitative classes describing the dominant type of falling particles in a given volume. Polarimetric weather radar can provide relevant information to discriminate particles regarding to their size, shape, phase state and orientation, and various hydrometeor classification algorithms have been proposed (e.g., Chandrasekar et al., 2013; Besic et al., 2016). These products are particularly powerful because they enable the sampling of a large spatial domain at a high temporal resolution. Even though some effort has been devoted to validation (Ryzhkov et al., 2005), one existing shortcoming of these methods lies in the difficulty of collecting in-situ information about the hydrometeor type. Direct measurements can be performed by airborne imaging probes but it remains a complex and expensive approach. Several studies have nonetheless addressed automatic hydrometeor classification from airborne particle images. Moss and Johnson (1994) classified images obtained with an Optical Array Probe (OAP) into seven categories using a boolean decision tree approach and Korolev and Sussman (2000) used an algorithm comparing dimensionless ratios of simple geometrical measures to discriminate between four families of snow particles. More recently, advanced pattern-classification algorithms such as artificial neural networks (Feind, 2006), and Principal Component Analysis (Lindqvist et al., 2012) were applied to achieve cloud particle classification with accuracies higher than 80%.

Another more accessible alternative to assess the performance and reliability of remote sensing retrievals is to compare the output with the hydrometeor types observed at ground level by in-situ measurement devices (e.g., Colle et al., 2014; Grazioli et al., 2015; Besic et al., 2016). For this purpose, ground-based snowflake imagers like the two-dimensional video disdrometer (2DVD, Kruger and Krajewski 2002), the Hydrometeor Velocity and Shape Detector (HVSD, Barthazy et al. 2004), the Snowflake Video Imager (SVI or PIP in its newest version, Newman et al. 2009) and the Multi-Angle Snowflake Camera (MASC, Garrett et al. 2012) provide relevant information in the form of two-dimensional binary or grayscale particle images and in some cases the associated fallspeed measurements. Recent investigations have shown the potential of the 2DVD to automatically detect and classify hydrometeors imaged according to their type and riming extent (Grazioli et al., 2014; Gavrilov et al., 2015). Bernauer et al. (2016) also proposed a decision-tree approach to distinguish between three degrees of riming by deriving constraints on the particle shape and fallspeed parameters measured by the 2DVD. However, the limited resolution of the device (about 0.2 mm) and the lack of information about the internal structure of the particles because of the binary nature of the images limited those studies to one minute averaged classification or binary graupel-snowflake classification scheme. Grayscale photographs bring additional information about the texture and surface roughness of the hydrometeors and therefore have the potential to evaluate the riming degree of individual particles (Nurzyńska et al., 2012). A recent investigation by Garrett and Yuter (2014) also showed that the fallspeed-size and fallspeed-shape relationships were highly uncertain at ground level, the measured fall velocities being strongly affected by local turbulence effects. Consequently, it is desirable to propose a hydrometeor classification algorithm which does not rely on any *a priori* knowledge of these relationships.

In this context, the information provided by the MASC is particularly relevant to classify individual hydrometeors and evaluate the extent of riming. The MASC is a ground-based snowflake imager which captures photographs of falling hydrometeors





from three different angles while measuring their fall velocity. The high-resolution ($\sim 33\mu$m per pixel) grayscale stereoscopic images allow the expert user to identify the observed particles individually. This allows for a supervised approach relying on a manually labeled training set in order to achieve hydrometeor classification. The Multinomial Logistic Regression (MLR) is a well-known and long-standing machine learning method (Bishop, 2006, p.$\sim$209) which is used to address supervised mul-

5 ticlass classification problems like pattern recognition and image classification. For instance, it has been applied recently for land-use land-cover classification using ancillary soil data (Kempen et al., 2009) and airborne hyperspectral images (Li et al., 2010). MLR is a probabilistic model that assigns to observations a probability of belonging to each class introduced in the model, based on a maximum likelihood estimator. The classification procedure relies on a set of numerical features calculated for each observation. In contrast to other machine learning methods like recurrent neural networks or support vector machine,

the MLR works directly in the original feature space and allows for a direct interpretation of the regression weights. Another advantage of the method lies in the probabilistic information which can be used to provide a degree of confidence associated with each prediction.

This article introduces a new method which makes use of MLR to automatically classify individual hydrometeors observed by a MASC (and potentially other imaging sensors), based on a large set of geometrical and texture-based features devel-

15 oped for this purpose. The paper is structured as follows: section 2 describes the experimental setup and the MASC image processing procedure. Section 3 presents the proposed classification model. The main results, a comparison with independent measurements as well as some classification examples are given in section 4. Finally, a conclusion summarising the work and presenting some future perspectives is drawn in section 5.

## 2 Data description

### 2.1 Data collection

Images used to develop and evaluate the hydrometeor classification were obtained with a MASC. The MASC data were collected during two different measurement campaigns organized during the winter 2015-2016. The first campaign took place from October 2015 to June 2016 in Davos, Switzerland. During that time, the MASC was deployed in a Double Fence Intercomparison Reference (DFIR) at a meteorological test site located at 2540 m a.s.l. Also present in the DFIR during the

25 measurement campaign were a 2DVD and a weather station. The second campaign took place from November 2015 until January 2016 on the Antarctic French base of Dumont d'Urville, in the framework of the Antarctic Precipitation, Remote Sensing from Surface and Space project (APRES3, http://apres3.osug.fr). Collocated measurements from a weighing precipitation gage and a weather station were also collected. In total, more than two million of MASC images were collected and processed during these two measurement campaigns.



## 2.2 MASC instrument

The MASC is a ground-based instrument which automatically takes high resolution and stereoscopic photographs of hydrometeors in free fall while measuring their fall velocity. Its working mechanism being extensively explained in Garrett et al. (2012), we will only recall the main aspects here. The imaging unit is composed of three high resolution cameras attached to a ring structure and separated by an angle of 36°. Each camera points at an identical focal point lying in the middle of the ring structure, at approximatively 10 cm from the cameras. The triggering unit is composed of two pairs of horizontally aligned near-infrared emitter-receiver arrays, delimiting a measuring cross-section of approximatively 2.5 cm$^2$. Particles falling successively through both arrays are detected and trigger the three cameras as well as three spotlights used to illuminate the target. The two MASCs used in the present study were using identical 2448x2048 pixels cameras mounted with 12.5 mm lenses. The cameras' aperture and exposure time were adjusted in order to maximize the contrast on hydrometeor photographs while preventing motion blur effects. With these settings, in-focus image resolution was measured to be about 33 $\mu$m per pixel using a graduated calibration target.

## 2.3 Image processing and feature extraction

Similar to the human brain, a computer algorithm requires a set of criteria to rely upon for image classification. In the present case, this set of criteria takes the form of numerical descriptors, commonly called features in machine learning studies and computed from the particle photographs. Regardless of the classification method used, extracting an exhaustive and relevant set of features and avoiding redundancy are two essential steps as they will strongly affect the performance of the classifier. Because it is *a priori* impossible to know exactly what features are relevant to the target concept (i.e. hydrometeor classification), a large set of 72 descriptors derived from the particle size, shape, and textural information was introduced. Several of them have already been used for hydrometeor identification purposes in previous works (e.g., Lindqvist et al., 2012; Nurzyńska et al., 2012; Grazioli et al., 2014; Schmitt and Heymsfield, 2014). As we experienced some issues with the MASC fallspeed measuring unit during the campaign in Davos, this parameter was discarded in the proposed methodology.

Some of the descriptors extracted being highly correlated, a feature selection method is applied (Sec. 3.5) in order to avoid redundancy and reduce the dimensionality of the problem. The complete list of descriptors introduced is displayed in Table A1 in appendix, and can be divided into 7 categories. For the sake of brevity, we only give here a general description for each of those categories. It is important to note that only the two last families of features make use of the textural information provided by grayscale images. As a result, the first 59 descriptors displayed in Table A1 can be calculated from binary silhouette only. The analysis of MASC images and the feature extraction procedure were conducted using MATLAB Image Processing Toolbox Release R2015b.





### 2.3.1 Particle size and area

This subset contains features directly related to the size of the particle photographed like the projected area (with or without taking into account holes) and the perimeter, as displayed in Fig.1. Three different diameters (maximal, average and equivalent-area) are calculated, following the methodology of Hogan et al. (2012).

### 2.3.2 Elliptical approximations

In order to get some insight into the shape and aspect ratio of the hydrometeors, an ellipse is fitted to the outline of the particle in a least squares sense. The particle morphology is further described by calculating two more ellipses: the smallest circumscribed and the largest inscribed ellipse having the same center and orientation as the initial fit, as illustrated in Fig.1. The axes of each ellipse are determined using a heuristic optimization technique maximizing (respectively minimizing) their area. The descriptors extracted from these elliptical approximations are the ellipses' parameters (minor/major axis length, perimeter, area, aspect ratio) as well as some proportionality ratios between them (e.g. inscribed/circumscribed ellipse area ratio).

### 2.3.3 Particle shape

The falling hydrometeors observed with the MASC frequently exhibit typical geometrical patterns (e.g. rectangular for columnar crystals, hexagonal for planar crystals, spherical or conical for graupels). In order to detect these features, the particle bounding rectangle and smallest encompassing circle are computed, as illustrated in Fig.1. The calculation is made by using a heuristic algorithm which first determines the smallest convex set of points that contains the particle, also called convex hull. The descriptors thus extracted are essentially ratios between the particle area and the area of the calculated shapes. In addition, the particle complexity introduced by Garrett and Yuter (2014) is used as an input feature here. Finally, the hydrometeor fractal dimension calculated with a box-counting method (see for example Sarkar and Chaudhuri, 1994) is added to the feature list as complementary information on the particle shape complexity.

### 2.3.4 Morphological skeleton

The digital morphological skeleton is a single-pixel-width pattern condensing the information of an original binary silhouette without changing its connectedness and computed by means of morphological operators (Soille, 2013, p.152). In this study, the skeleton (illustrated in Fig.2) was obtained by successively applying discrete morphological closing, opening and thinning operations. The two first operations are performed in order to smooth the target outline and avoid the generation of undesired short end branches during the skeletonization. The number of junctions and ends included in the skeleton as well as the ratio between its length and the initial perimeter of the particle are examples of descriptors computed out of this procedure.



### 2.3.5 Rotational symmetry

Many observed snow crystals display a rotational symmetry. In order to highlight this symmetry, a few descriptors based on the radial distance between the snowflake outline and its centroid have been implemented. First, the angular outline-to-centroid distance is computed and discretized into 360 bins (1 degree per bin). The resulting vector of distances is then normalized to

have 0 mean and 1 standard deviation. Finally, a Fourier transform is performed on the normalized signal and the components 0 to 6 of the resulting Fourier power spectrum are used as input features for hydrometeor classification. For example, plates and dendrites are characterized by a larger value of the sixth component of the power spectrum than other hydrometeors, as illustrated in Fig.2. Additionally, the mean and standard deviation of the non-normalized angular outline-to-centroid distance are added to the list of descriptors.

### 2.3.6 Texture operators

The textural information provided by the grayscale photographs from the MASC has been processed to extract global features based on the pixel intensity distribution computed over the whole particle silhouette. Those include snowflake average and maximum brightness, brightness standard deviation, image contrast and histogram entropy. In addition, several local operators commonly used in computer vision tasks have been adapted to MASC images: the gray-level mean local variance, the energy

of Laplacian and the sum of wavelet coefficients, as detailed in Pertuz et al. (2013).

### 2.3.7 Co-occurrence matrix

The gray-level co-occurrence matrix (GLCM) $C$ is a measure of the distribution of co-occuring pixels at a given spatial offset characterized by a distance $d$ and an angle $\theta$:

$$C_{d,\theta}(p,q) = \sum_i \sum_j \begin{cases} 1, & \text{if } I(i,j) = p \text{ and } I(i \pm |d\cos\theta|, j \pm |d\sin\theta|) = q, \\ 0, & \text{otherwise,} \end{cases} \tag{1}$$

where $p, q$ are pixel values and $i, j$ their spatial position within the image $I$. The GLCM therefore provides information about the spatial relationship between different pixel intensities, and more generally about the spatial structure of the imaged hydrometeor. The so-called Haralick features are an ensemble of statistical operators which attempt to summarize the information contained in the GLCM in 14 numerical descriptors (Haralick et al., 1973). In the present study, four Haralick features (angular second moment, contrast, correlation and homogeneity) have been computed using a distance offset $d = 1$ and along 4 princi-

pal directions characterized by $\theta \in \{0°, 45°, 90°, 135°\}$, following the methodology of Eleyan and Demirel (2011). In an effort to extract isotropic textural descriptors which do not depend on the spatial orientation of the imaged snowflake, the Haralick features were eventually averaged over the four directions before being used for the classification task.



## 3 Hydrometeor classification

This section details the proposed classification methodology. As the imaged hydrometeors can be identified by human inspection, a supervised approach was preferred. Supervised classification aims to learn the association between input features and output classes from a labeled dataset called the training set and to generalize to unlabeled observations. The requirements for supervised classification are the definition of a classification scheme, the selection and labeling of a training set as well as the implementation of a classification method. In this section. we explain these three steps successively.

### 3.1 Classes definition

Multiple solid hydrometeor classification schemes have been introduced in the literature (e.g., Korolev and Sussman, 2000; Lindqvist et al., 2012; Grazioli et al., 2014) and it is often difficult to find an equivalence between them, partly because the type of particles probed depend on the observation conditions (ground-based or airborne instrument, measurement location). In the present work, efforts were made in order to develop an exhaustive and versatile classification scheme applying the following procedure. First, we started from the nine main snow habit categories introduced by Magono and Lee (1966). In a second step, the classes Aggregate (particles resulting from the combination of two or more colliding crystals) and Small Particle were added to the model. For the sake of simplicity, Needle and Column having very similar geometrical properties, they were merged into a single class called Columnar Crystal. The resulting classification scheme composed of ten classes is the following: Small Particle, Columnar Crystal, Planar Crystal, Combination of Columnar Crystals, Combination of Planar Crystals, Combination of Columnar and Planar Crystals, Aggregates, Graupels, Irregular Snow Crystal, Germ of Snow.

The MASC being a ground instrument, some of the snow classes were hardly ever observed in the data collected. It was therefore not possible to include enough samples in the training set to perform a reliable automatic classification. The classes concerned are Combination of Columnar Crystals (e.g. bullets rosette), Combination of Planar Crystals (e.g. radiative assemblage of plates), Irregular Snow Crystal and Germ of Snow Crystal. It should be noted that they could easily be reintroduced in future classification studies based on different datasets that include these types. Finally, the ice-phase hydrometeor type classification scheme utilized in the present work is illustrated in Fig.3**a** and is composed of the six following classes: Small Particle (SP), Columnar Crystal (CC), Planar Crystal (PC), Combination of Columnar and Planar Crystals (CPC), Aggregate (AG) and Graupel (GR). In the rest of this paper, we will refer to this classification scheme as hydrometeor type.

The high resolution photographs of the MASC are detailed enough to observe and quantify the presence of cloud frozen droplets on the surface of the particles. In addition to the hydrometeor type, a continuous riming index $\mathcal{R}_c$ lying between 0 (no riming observed) and 1 (graupel) is hence introduced. Five distinct qualitative degrees of riming adapted from Mosimann et al. (1994) and detailed in Table 1 have been selected and mapped to [0,1] using a sinusoidal function. If we denote by $\mathcal{R}_d \in [1,5]$ the qualitative degree of riming, the mapping function applied is:

$$\mathcal{R}_c = \frac{1}{2} \left( \sin \left( \frac{\pi}{4} (\mathcal{R}_d - 3) \right) + 1 \right). \tag{2}$$

The purpose of the sinusoidal transformation is to better reflect the non-linear spacing between each value of $\mathcal{R}_d$ as well as to increase the sensitivity in the middle of the scale, i.e. around *densely rimed* particles. Indeed, the riming process required to





transit from degree 2 to 3 and 3 to 4 is much more intense than between 1 and 2 or 4 and 5. Figure 3**b** summarizes the resulting riming index scale, which is used to automatically quantify the presence of riming.

Melting falling snow is characterized by eroded particle outlines as well as the presence of liquid water droplets forming on their surface. On MASC images, these liquid water droplets appear as glints of reflection identified by their small size and saturated pixel values. It is therefore possible to detect melting snowflakes using the geometrical and textural descriptors detailed in section 2.3. The detection is achieved through a binary classification between *dry* and *melting* hydrometeors as illustrated on Fig.3**c**. Because the reflection in liquid water is small and practically independent from the size of the drop, it is very difficult to differentiate raindrops from small particles on the basis of the image only. As a result, a liquid precipitation event will be identified by a proportion of small particles close to 100%.

## 3.2 Training set

The preparation and labeling of a training set is a very important step as it strongly influences the learning phase and directly impacts the ability of the model to generalize to unknown data. For the present work, a dataset of $N_{\text{label}} = 3556$ MASC particle images was selected in an effort to reflect the proportions between the hydrometeor classes, as observed during the campaigns. In a second step, four operators independently labeled the training set according to the three classification schemes introduced in section 3.1 (i.e. hydrometeor type, riming degree and dry/melting snow), with the possibility to flag certain samples as undetermined (for instance, it is difficult to assess the degree of riming of a melting snowflake). For each classification scheme, only the particles labeled identically by at least 3 operators have been retained for training the classifier. This resulted in a total of $N_{\text{label}}^1 = 3238$ images for training the hydrometeor type model and $N_{\text{label}}^2 = 3183$ for the riming degree. As the original training set contained too few samples of melting snowflakes, it has been extended with more than 1500 images observed during wet snow events ($N_{\text{label}}^3 = 4987$).

## 3.3 Classification method

### 3.3.1 Multinomial logistic regression

Logistic regression is a statistical probabilistic model used to perform binary classification of a dependent variable $y \in \{0,1\}$ based on the value of $D$ input variables gathered in a vector $\mathbf{x} \in \mathbb{R}^D$. As a member of the generalized linear model family, logistic regression can be seen as an analogue to the linear regression in which the linear model $\tilde{\mathbf{x}}^T \boldsymbol{\beta}$ is passed through a logistic function $\sigma(x) = \frac{\exp(x)}{1+\exp(x)}$. With this notation, $\tilde{\mathbf{x}} = (1, \mathbf{x}^T)^T$ is the augmented features vector and $\boldsymbol{\beta} = (\beta_0, \ldots, \beta_D)^T$ is the vector of linear regression weights.

Multinomial logistic regression is an extension of this model for the case where $y$ has more than two categorical outcomes, ie. $y \in \{1, \ldots, K\}$ where $K$ denotes the number of classes. In this case, the logistic function is generalized to a *softmax* function





and the probability of having an instance $y_n = k$ among $K$ classes is given by

$$p\left(y_n = k \mid \tilde{\mathbf{x}}_n, \mathbf{B}\right) = \frac{\exp\left(\tilde{\mathbf{x}}_n^T \boldsymbol{\beta}_k\right)}{\sum_{j=1}^{K} \exp\left(\tilde{\mathbf{x}}_n^T \boldsymbol{\beta}_j\right)}, \tag{3}$$

where $\mathbf{B} = \{\boldsymbol{\beta}_1, \ldots, \boldsymbol{\beta}_K\}$ is the matrix of linear regression weights. The goal now is to estimate the model parameters contained in $\mathbf{B}$ which best explain the observed pairs of data $\{y_n, \mathbf{x_n}\}_{n=1}^{N_{\text{train}}}$ comprised in the training set. In the Bayesian formulation, this can be done by maximizing the likelihood of observing the data $\mathbf{y}$ given $\tilde{\mathbf{X}}$ and $\mathbf{B}$. Assuming that each $y_n$ is independent of all other elements of $\mathbf{y}$, one can write:

$$\mathbf{B}_{lik} = \operatorname*{argmax}_{\mathbf{B}} p(\mathbf{y} \mid \tilde{\mathbf{X}}, \mathbf{B}) = \operatorname*{argmax}_{\mathbf{B}} \prod_{n=1}^{N} p(y_n \mid \tilde{\mathbf{x}}_n, \mathbf{B}). \tag{4}$$

Similarly to the least squares error for linear regression, one can build a cost function $\mathcal{C}_1(\mathbf{B})$ by taking the negative logarithm of the likelihood introduced in (4):

$$\mathcal{C}_1(\mathbf{B}) = -\sum_{n=1}^{N} \sum_{k=1}^{K} \tilde{y}_{nk} \tilde{\mathbf{x}}_n^T \boldsymbol{\beta}_k + \sum_{n=1}^{N} \log \sum_{j=1}^{K} \exp\left(\tilde{\mathbf{x}}_n^T \boldsymbol{\beta}_j\right), \tag{5}$$

with $\tilde{y}_{nk} = 1$ if $y_n = k$ and 0 otherwise. The optimal solution of (5) is then found by minimizing the cost function using the Newton-Raphson method. Once $\mathbf{B}$ is calculated, the probability for an unknown sample $\mathbf{x}_u$ of belonging to the class $k$ is computed using (3). The class which obtained the highest probability is then assigned to the sample.

### 3.3.2 Regularized MLR

In order to reduce the dependence of the final model on training samples and to avoid over-fitting, it is common to add a regularization term to the cost function introduced in section 3.3.1. From a Bayesian point of view, this is equivalent to adding a prior distribution for the $\{\boldsymbol{\beta}_k\}$. In this study, the simplest approach which is to impose a univariate Gaussian prior with mean 0 and variance $\sigma^2$ is adopted (also known as L2 regularization):

$$p(\boldsymbol{\beta}_k \mid \sigma^2) = \frac{1}{\sqrt{2\pi\sigma^2}} \exp\left(-\frac{\boldsymbol{\beta}_k^2}{2\sigma^2}\right). \tag{6}$$

Assuming that the $\{\boldsymbol{\beta}_k\}$ are independent from each other and hence the prior for $\mathbf{B}$ is the product of the priors for each $\boldsymbol{\beta}_k$, one can reformulate the maximum likelihood estimate (Eq. 4) into a maximum *a posteriori* estimate by including the prior for $\mathbf{B}$ in the equation:

$$\mathbf{B}_{map} = \operatorname*{argmax}_{\mathbf{B}} p(\mathbf{y} \mid \tilde{\mathbf{X}}, \mathbf{B}) p(\mathbf{B} \mid \sigma^2) = \operatorname*{argmax}_{\mathbf{B}} \prod_{n=1}^{N} p(y_n \mid \tilde{\mathbf{x}}_n, \mathbf{B}) \prod_{k=1}^{K} p(\boldsymbol{\beta}_k \mid \sigma^2). \tag{7}$$

Following the same procedure as in section 3.3.1, one can derive the equivalent regularized cost function $\mathcal{C}_2$:

$$\mathcal{C}_2(\mathbf{B}) = -\sum_{n=1}^{N} \sum_{k=1}^{K} \tilde{y}_{nk} \tilde{\mathbf{x}}_n^T \boldsymbol{\beta}_k + \sum_{n=1}^{N} \log \sum_{j=1}^{K} \exp\left(\tilde{\mathbf{x}}_n^T \boldsymbol{\beta}_j\right) + \lambda \sum_{j=1}^{K} \boldsymbol{\beta}_j^T \boldsymbol{\beta}_j, \tag{8}$$





where $\lambda = \frac{1}{2\sigma^2}$ is a hyperparameter controlling the degree of regularization. The higher the value of $\lambda$ is, the more we favor smaller values for $\boldsymbol{\beta}_k$. In that sense, this new cost function prevents the model from over-fitting the training set by choosing arbitrarily large values for the $\{\boldsymbol{\beta}_k\}$.

### 3.3.3 Cost adjusted regularized MLR

In the presence of imbalanced dataset (i.e. significantly different numbers of data samples between certain classes), standard classification methods such as MLR are often biased towards the majority class(es). As it was pointed out in section 3.2, the training set is composed of imbalanced class distributions. Indeed, some classes like Combination of Columnar and Planar Crystals for the hydrometeor type or riming degree $= 0$ were rarely observed during the measurement campaigns. Even though this issue was somewhat mitigated during the selection of the training set, the disproportion is still present in the data. In the

recent literature, several techniques have been proposed to address this problem (e.g., He and Garcia, 2009; López et al., 2013). In the present work, we applied a simple cost-sensitive learning method which consists of weighting each training sample in the cost function by a factor inversely proportional to its occurrence frequency in the training set. If we denote by $f_n$ the proportion of data belonging to the same class as $y_n$, the final cost function utilized in this study can be written as:

$$\mathcal{C}_3(\mathbf{B}) = -\sum_{n=1}^{N} \omega_n \sum_{k=1}^{K} \tilde{y}_{nk} \tilde{\mathbf{x}}_n^T \boldsymbol{\beta}_k + \sum_{n=1}^{N} \omega_n \log \sum_{j=1}^{K} \exp\left(\tilde{\mathbf{x}}_n^T \boldsymbol{\beta}_j\right) + \lambda \sum_{j=1}^{K} \boldsymbol{\beta}_j^T \boldsymbol{\beta}_j, \tag{9}$$

where $\omega_n = 1/(K f_n)$. One can note that the regularization term is not affected.

### 3.4 Feature transformation

In machine learning, it is common practice to apply some transformations to the input variables to better meet the assumptions of the chosen classification method. In the case of MLR, the input variables do not necessarily need to follow a multivariate normal distribution, although normality generally yields to a better and more stable solution whilst limiting the impact of out-

liers. In order to better fulfill this condition, the descriptors introduced in Sec. 2.3 were transformed based on their distribution skewness in the training data. If we denote by $\mathbf{x}_d \in \mathbb{R}^N$ the distribution of the $d$th feature in the training data and by $\mathcal{S}_d$ its skewness, then the following transformation was applied:

$$\mathbf{x}_d = \begin{cases} \exp(\mathbf{x}_d) & \text{if } \mathcal{S}_d < -1, \\ \mathbf{x}_d^2 & \text{if } -1 < \mathcal{S}_d < -0.5, \\ \sqrt{\mathbf{x}_d} & \text{if } 0.5 < \mathcal{S}_d < 1, \\ \log(\mathbf{x}_d) & \text{if } \mathcal{S}_d > 1. \end{cases} \tag{10}$$

This data treatment significantly increased the classification accuracy. In order to deal with feature distributions ranging

across different scales, the resulting distributions were further normalized to 0 mean and 1 variance before training the classification algorithm.





### 3.5 Feature selection

Some of the 72 features developed in section 2.3 are redundant, with some being highly correlated (e.g. size related descriptors). The objective is therefore to remove as many redundant or irrelevant descriptors without decreasing the classification accuracy. A well-established and efficient approach to achieve this goal is to use feature extraction techniques such as Principle
Component Analysis (PCA) or Linear Discriminant Analysis (LDA) and then perform dimensionality reduction by removing the least important components (Jolliffe, 2002). For instance, PCA has been successfully applied for ice-cloud particle habit classification by Lindqvist et al. (2012). However, such techniques map the original descriptors into a new feature space, making the analysis and interpretation of the obtained features difficult. There is indeed not necessarily a physical meaning in the features obtained from a PCA. For this study, a greedy forward feature selection algorithm which does not modify the original
descriptors was implemented. The method works as follows: the search begins with an empty set of features and descriptors are iteratively added to the set. At each iteration, the method tests the classification performance obtained with the addition of every remaining descriptor separately, and the set which led to the best classification performance (in term of Heidke Skill Score, defined in Eq.13) is selected as a starting point for the next iteration. A more comprehensive explanation of the algorithm can be found in Tang et al. (2014).

### 3.6 Application to MASC data

The MASC instrument provides stereoscopic photographs from three different views for each hydrometeor detected. First, a particle detection and triplet matching algorithm is applied in order to extract at most one triplet of images cropped around the same snowflake. In a second step, descriptors are processed for each of the three views. Each image is then classified according to the three schemes introduced in Sec. 3.1 separately and independently from the two other views. Finally, the probabilistic
class membership index provided by the model (see Eq.3) is weighted on the three views in order to assign a final label to the hydrometeor. If we denote by $p_{ij}$ the probability given by the MLR model that the hydrometeor projection observed on camera $i$ belongs to the class $j$, then the label assigned to the image triplet is:

$$\text{label} = \underset{j}{\arg\max} \sum_{i=1}^{3} p_{ij}. \tag{11}$$

The methodology is slightly different for the evaluation of riming for which a continuous riming index is associated with
each view. In that case, the global riming index of the triplet is simply calculated as the average of the three obtained values.

In this manner, the whole procedure is versatile and can be easily adapted to other devices providing only a single or two (possibly binary) images per particle. A summary of the image processing and classification procedure is illustrated in Fig.4. Note that the three classifications are achieved in parallel hence no information is exchanged between them.

### 3.7 Performance assessment

The evaluation of the classification performance is conducted using different metrics. The labeled dataset $N_{\text{label}}$ is split into an effective training set $N_{\text{train}}$ used to fit the model parameters and a test set $N_{\text{test}}$ employed to test and validate the classification



on unused data. In order to use each data sample both for training and testing the method, 4-fold cross-validation is conducted. For each classification scheme, the accuracy of the method is assessed through the analysis of the confusion matrix between model predictions and real labels evaluated on test data, as well as by the introduction of three additional performance indices: the overall accuracy (OA), the Heidke skill score (HSS, also known as Cohen's kappa) and the balanced error rate (BER). If

we denote by $\mathbf{M}$ the confusion matrix, then $M_{ij}$ contained the number of test samples labeled in the $j$th class and predicted belonging to the $i$th class. Formally, the OA, HSS and BER are defined as:

$$OA = \frac{\sum_{i=1}^{K} M_{ii}}{N} \times 100, \tag{12}$$

$$HSS = \frac{OA - E}{1 - E}, \tag{13}$$

$$BER = \frac{1}{K} \sum_{k=1}^{K} \left[ \frac{1}{N_k} \sum_{n=1}^{N} \{y_n = k\}\{y_n \neq \hat{y}_n\} \right], \tag{14}$$

where $K$ is the number of classes, $N$ the total number of samples considered, $N_k$ the number of samples in class $k$, $y_n$ the true label associated with sample $n$ and $\hat{y}_n$ the predicted class. The term E appearing in the HSS evaluates the number of correct predictions that could occur by chance and is computed from the confusion matrix as:

$$E = \frac{1}{N^2} \sum_{i=1}^{K} M_{i,*} M_{*,j}, \tag{15}$$

with $M_{i,*}$ the total for the $i$th row and $M_{*,j}$ the total for the $j$th column. In order to assess the increase in performance brought

by the MLR method, a simple baseline model was built. It consists of the most simple yet relevant way of classifying new samples given an ensemble of $D$ features and a training set. The model simply assigns a given particle to the class $k$ whose centroid is the closest in the feature space. Hence for a given sample $i$:

$$k_i = \underset{k}{\mathrm{argmin}} \sum_{j=1}^{D} \left( x_{ij} - \frac{1}{N_k} \sum_{m \in k} x_{mj} \right)^2, \tag{16}$$

where $x_{ij}$ is the value of descriptor $j$ for particle $i$ and $\frac{1}{N_k} \sum_{m \in k} x_{mj}$ is the mean of the feature $j$ among all the training

samples belonging to the class $k$.

## 4   Results

This section is divided into five parts. In the three first subsections, the classification performances obtained for each classifier (i.e. hydrometeor type, riming degree, melting snow detection) are reported and analyzed independently. The overall information is eventually merged in subsection four and a few examples of application to MASC images triplet are presented. Finally,

a comparison with an existing classification method applied to measurements collected in the Swiss Alps with a collocated 2DVD is conducted and illustrated through two precipitation events in subsection five.



## 4.1 Hydrometeor type

The performances of the hydrometeor type classification model were evaluated on the basis of the $N_{\text{label}}^1 = 3238$ data samples manually labeled by human inspection. The initial set of 72 features was first reduced in order to get rid of the irrelevant or redundant descriptors. This was achieved using the greedy forward feature selection algorithm presented in Sec.3.5. For each subset of features, 4-fold cross-validation was conducted and the average HSS evaluated on test data was calculated. The procedure was repeated 10 times based on different random splitting between training and test set in order to get insight into the statistical fluctuations around the HSS values obtained. The results are displayed in Fig.5 and show that the 10 descriptors added first significantly improve the classification accuracy. The curve then flattens and seems to reach a plateau around ∼25-30, meaning that the remaining features are not relevant to the classification task. For the sake of readability, the figure was cut after 50 features. For this study, we decided to keep only the first 25 features, flagged with a "I" in Table A1.

The classification accuracy was then assessed by carrying out 10 iterations of 4-fold cross-validation. The degree of regularization of the model, controlled by the hyper parameter $\lambda$ introduced in Eq.8, was tuned to maximize the test HSS following a grid-search procedure. Figure 6 shows the confusion matrix based on test data and averaged over the 10 iterations of cross-validation. In order to better visualize the unbalanced proportions between certain classes, the matrix is normalized to sum up to 100 hence each field represents percentage. The classifier performs well with 94.7% of the test samples located on the diagonal. The accuracy is further evaluated with the OA, HSS and BER. Table 2 shows the scores obtained by the MLR method as well as by the baseline model. The high value of the average HSS, similar to the OA, indicates a high classification accuracy and a low probability of correct classification occurring by chance. The average BER is below 7% and its standard deviation, computed over 40 instances of test error, is equal to 1.1%. The mean error rate per class is contained between 1% and 15% and varies as follows: SP 1.3%, CC 3.3%, PC 12.5%, CPC 15.0%, AG 5.2% and GR 1.8%. The higher error rate for CPC might be explained by the low occurrence of this class in the collected data, resulting in a small proportion in the training dataset (i.e. 120 samples). In the future, this could be improved by extending the training data with additional samples. For PC, the relatively high error rate is probably due to the difficulty in distinguishing them from aggregates of a few stellar crystals. This ambiguity is also present in the confusion matrix (Fig.6) with a significant number of PC being classified as AG and vice-versa.

The baseline model obtained a mean BER of 15.2% and a mean HSS of 0.81 and therefore confirmed that the computed features are discriminating well between the classes and are relevant to the target concept. The difference in HSS between the two models is of 0.12 and can be interpreted as the added value brought by the MLR method.

One convenient property of MLR is that the model is linear, in the sense that it works directly in the original feature space without any remapping. As a result, each $\beta_d$ weight introduced in section 3.3.1 can be directly related to the $d$th descriptor. Recalling that every feature is normalized to have the same mean and variance, the absolute value of those weights can further be used to assess the importance of the associated descriptor in the model. As there is one value of $\beta_d$ per class defined, the average value over each class $\bar{\beta}_d$ was used as an indicator of the importance of feature $d$ in the model. Figure 7**a** displays the results thus obtained. The 5 features having the largest weight in the logistic regression are, in decreasing order: Haralick correlation, Distance to centroid standard deviation over mean ratio, Particle/convex hull area ratio, $D_{\text{max}}$, Inscribed ellipse/fitted





ellipse area ratio. This does not imply that the other features are irrelevant or negligible, but that there is potentially a more immediate relationship between the model target (i.e. hydrometeor type) and the top-ranked features. It is interesting to see that among the 25 features kept, only 7 make use of the textural information given by the grayscale images. We attempted to simulate the results that we would obtain using a snowflake imaging device providing binary images of similar resolution by

removing the texture-related features. By doing so, we obtained a HSS of 0.9 and a BER of 8.9%, hence indicating a very good performance using only geometric information.

Finally, the completeness of the training set as well as the stability of the classifier were evaluated by computing learning curves. First, 25% of the labeled data were held apart and kept as a fixed test dataset. Then, the size of the training set was iteratively increased from 2% to 98% of the rest of the labeled dataset. For each considered proportion, 20 iterations of random

training data sampling were performed, a classification was conducted and the train and test errors were recorded. The results obtained from this procedure are presented on Fig.8 in the form of two learning curves based on the BER. The test error curve quickly decreases at first (when the size of the training set is small) and reaches a plateau around 60% of training data used, which corresponds to ~1460 images. This suggests that the number of labeled images (3238) available for the present study is sufficient. Furthermore, the converging behavior of the train and test learning curves indicates that the model is not sensitive to

training data sampling and generalizes well to unknown data.

## 4.2   Riming degree

The presence of riming categorized into 5 qualitative classes was learned by the model based on the $N_{label}^2 = 3183$ labeled training data. First, the greedy forward feature selection method detailed in Sec.3.5 was applied in order to discard the descriptors that are irrelevant for the estimation of the degree of riming, as illustrated on Fig.5. For the sake of consistency, the

25 features firstly added were kept for the classification task and flagged with a "II" in Table A1. The classification accuracy was assessed on test data by conducting 10 iterations of 4-fold cross-validation and the scores obtained are reported in Table 2. Even though these values are significantly worse than for the hydrometeor type model, one can see on the associated confusion matrix (Fig.9a) that more than 95% of the misclassification is located next to the diagonal. This pattern indicates that the classifier discriminates well between non-rimed, rimed and graupel particles but predictions come with an uncertainty of

±1 level. Recalling that the riming degree is a continuous value that was categorized into 5 qualitative levels for the sake of simplicity, the associated riming index $\mathcal{R}_c$ (see Eq.2) was weighted by the probabilistic output of the MLR model in order to get a continuous value in $[0, 1]$. The results are illustrated on Fig.9b and show a good agreement between the median values of $\mathcal{R}_c$ obtained and their expected value (i.e. 0, 0.15, 0.5, 0.85, 1 for $\mathcal{R}_d = 1, 2, 3, 4, 5$, respectively). The largest bias is obtained for densely rimed particles ($\mathcal{R}_d = 3$) with a value of $-0.08$, indicating that the classifier tends to underestimate the riming

index for this class. In order to quantify the uncertainty around the predictions, we evaluated the difference between the labeled riming index and the predicted one, resulting in the histogram showed in Fig.10. The displayed distribution is characterized by a mean $\mu \simeq 0.00$, a standard deviation $\sigma \simeq 0.13$ and exhibits strong symmetrical properties. The riming index associated with each hydrometeor seems therefore to be generally well predicted and characterized by a probable error of 5.3%, calculated as the semi-interquartile range over the full range of the error.



Similarly to what was done in section 4.1 for the hydrometeor type, the 25 utilized descriptors were classified by order of importance in the model and separated between geometrical and textural descriptors, as displayed on Fig.7**b**. Interestingly, the 5 top-ranked features are clearly defined with a significantly higher weight. Moreover, they are all making use of the textural information, suggesting that the presence of riming affects primary the surface roughness and light reflection properties of

5 the particles. When the riming process is intense enough, a particle's projected area tends to expand and smooth its initial outline, which will be detected by the geometrical descriptors as well. When discarding the textural descriptors and running the classification again, we obtained a HSS of 0.50, 25% lower than initially. We can therefore conclude that the added value brought by the grayscale images is especially useful to detect and quantify the presence of riming.

### 4.3 Melting snow

Binary logistic regression was applied to detect melting snow and classify MASC images between *dry* and *melting* hydrometeors on the basis of $N_{\text{label}}^3 = 4987$ labeled images. The dataset contains 1636 samples of melting snow and is principally composed of melting aggregates. In a first step, greedy forward feature selection was performed and allowed us to select 25 features kept for the classification (Fig.5), following the same procedure as for the hydrometeor type and degree of riming (see Sections 4.1 and 4.2). The features selection procedure is illustrated on Fig.5**c** and the 25 chosen descriptors are flagged with

a "III" in Table A1 in Appendix.

In a second step, classification performances were assessed by conducting 10 iterations of 4-fold cross-validation. The mean scores obtained and the associated confusion matrix based on test data are reported in Table 2 and Fig.11, respectively. The classification accuracy is characterized by a HSS of 0.86, an OA of 93.9% and a BER of 7.1%. The confusion matrix indicates a very good performance with a false positive rate of detecting melting snow of 4.4%.

Finally, features were rearranged by order of estimated importance following the same procedure as for the hydrometeor type and riming degree classifications. The resulting feature weights are displayed on Fig.7**c**. As for the riming degree, it seems that the textural features computed from grayscale images have a significant importance for the classification, 3 of them being present in the 5 top-ranked descriptors. The HSS obtained after removal of the 8 textural descriptors is 0.76, 12% lower than initially.

Learning curves similar to the ones obtained for the hydrometeor type classification (Fig.8) were also computed for the riming degree and melting snow detection in order to assess the quality of the training set and the stability of the classifier. The associated figures are not reported here because they are very similar and lead to the same conclusions.

### 4.4 Application to unlabeled MASC data

The information obtained from the hydrometeor type classification, the riming degree evaluation and the detection of melting

snow has to be merged in order to assign a unique label to each image triplet provided by the MASC. This is done following the methodology explained in section 3.6. Figure 12 illustrates the procedure and the predictions obtained for two unlabeled MASC triplets. On panel **a**), one can see a dry, heavily rimed aggregate. On the two first views, the model recognized it as an aggregate, even though it also detected some aspects belonging to the planar crystal class. On the third view, the snowflake



projected area exhibits an aspect ratio close to one. The particle features a simpler outline, some 6-fold symmetry and a surface roughness typical from graupel-like snowflakes. As a result, the model prediction is more uncertain. By merging the predictions obtained on the three views, the model finally classified the triplet as a dry, heavily rimed ($\mathcal{R}_c = 0.68$) aggregate.

Panel **b**) exhibits a planar crystal falling with a canting angle of $\sim 35°$ and illustrates well the added value brought by MASC stereoscopic photographs for hydrometeor classification. On the two first views, the crystal shows a 6-fold symmetry and is classified as a planar crystal. However, the particle is seen edge-on in the third view and is therefore plainly classified as a columnar crystal. The final label assigned to this hydrometeor is a dry, moderately rimed ($\mathcal{R}_c = 0.24$) planar crystal, even though some characteristics of melting snow were detected on the first view.

In general, the image triplets provided by the MASC allowed the classifier to give more accurate predictions and, most importantly, to better characterize the associated uncertainty.

## 4.5 Comparison with an existing classification method

A comparison was conducted with respect to a hydrometeor classification scheme developed for 2DVD data and detailed in Grazioli et al. (2014). The method provides an estimate of the dominant type of hydrometeor measured by the instrument on a time interval of one minute. The classification is achieved by means of a Support Vector Machine employing the one minute statistical distribution of several descriptors as input features, and is therefore completely independent from the approach proposed in the present work. The class attribution was made according to the following classification scheme: Small particle-like (SP), Dendrite-like (D), Column-like (C), Graupel-like (GR), Rimed particle-like (RIM), Aggregate-like (AG), Melting snow-like (MS) and Rain (R). The "-like" was introduced to emphasize that the method only evaluates the dominant type of hydrometeor recorded within each time-step. In the following, we will call HC-2DVD, respectively HC-MASC the two methods compared.

The HC-2DVD algorithm was applied on data collected during the winter 2015-16 measurement campaign in Davos (CH). During that period, the two instruments were collocated in a DFIR sheltered from the wind, leading to an ideal configuration to compare the classification methods. In order to have enough measurements per bin and ensure a reliable comparison, the dominant hydrometeor type derived from MASC image triplets was calculated over time intervals of five minutes and HC-2DVD was adapted to the same time resolution. In a second step, HC-MASC and HC-2DVD classes were merged according to the following rules: 1) As the MASC detects raindrops as small particles, Rain and Small particle-like from the HC-2DVD scheme were merged into a unique class corresponding to Small particle from HC-MASC. 2) Dendrite-like (HC-2DVD) and Planar crystal (HC-MASC) were combined together. 3) Column-like (HC-2DVD) and Columnar crystal (HC-MASC) were combined together. 4) MASC time intervals composed of more than 30% of melting particles according to the binary dry/melting snow classifier were identified as Melting snow-like from the HC-2DVD method. This relatively low threshold was adjusted in order to best retrieve transitions between liquid and solid hydrometeors. Mixed-phase precipitation intervals are indeed composed by a large amount of small particles, which are not taken into consideration in the dry/wet snow classification. Over timesteps of five minutes, a higher threshold would lead to only very few intervals identified as melting snow and a reduced agreement



with HC-2DVD. 5) Rimed particle-like were assimilated to Aggregates-like, as crystal-dominant (both planar and columnar) five minutes time intervals were hardly ever observed during the campaign.

The comparison was first conducted over the whole campaign, from 12 October 2015 to 19 June 2016. For the sake of reliability, every five minutes interval containing at least 30 non-blurry MASC images and 300 2DVD particles were taken into consideration, resulting in a total of more than 88 hours of precipitation. The significantly higher threshold value chosen for the 2DVD results accounts for the instrument sampling area about 4 times larger than the MASC. MASC data were also pre-filtered in order to get rid of dark and/or blurry images. The comparison results are summarized in Table 3 in the form of a confusion matrix. In general, the two classification methods are in good agreement with a HSS of 0.56 and an OA of 72%. Aggregates was the most observed hydrometeor type during the campaign and is on average well detected by both methods. Some inconsistencies exist though, principally with the graupel class. The highest misclassification rate is found for time intervals classified as graupel by HC-2DVD and aggregates by HC-MASC. Manual observation of MASC images during some of those events revealed a majority of aggregates with various riming degree even though some graupels were present as well. The uncertainty associated with 2DVD classification (binary images, ∼10 times lower resolution) and the incapability to discern and discard blurry particles might explain this relatively high error rate. Numerous wet hydrometeors (rain, melting snow) time intervals are present in the dataset and indicate that both HC-MASC and HC-2DVD efficiently capture the transition between liquid-phase and ice-phase precipitation. For melting snow, the classification consistency is lower and show a large uncertainty with respect to the classes of aggregates and small particles/rain. This ambiguity might be coming from the numerous events recorded when the ground temperature was close to the freezing point. Under these conditions, both classifiers are recording intermittent moments of liquid, mixed and solid hydrometeors.

In the following, two precipitation events recorded by the MASC and 2DVD are presented and show time-series of HC-MASC and HC-2DVD co-fluctuation.

### 4.5.1 23 April 2016

The classification output of HC-MASC and HC-2DVD for a snowfall event occurring on 23 April 2016 is presented on Fig.13. The air temperature, as recorded by a weather station located in the DFIR, was below freezing and oscillating between -3 and -1°C. In both HC-MASC and HC-2DVD panels, the classifier displays a clear transition in the dominant type of hydrometeors around 08:30 UTC. Before 08:00 UTC, HC-2DVD exhibits several transitions between aggregates, rimed particles and dendritic crystals timesteps. This behavior is in good agreement with HC-MASC which shows a stable predominant proportion of aggregates. Moreover, the significant amount of pristine planar crystals detected could explain the few instants of dendritic crystals recorded by HC-2DVD. The riming index $\mathcal{R}_c$ recorded during this first phase was constant around 0.4, corresponding to densely rimed hydrometeors. From 08:30 UTC, the riming process intensifies within an hour, leading to a snowfall content composed by heavily rimed particles and graupels. This transition is also well retrieved by the riming index which progressively increases to ∼0.7. Further investigation on the development of the riming process which led to this transition is beyond the scope of the present study and would require additional information (e.g. vertical profile of the storm, values of the atmospheric variables aloft). Nevertheless, this example illustrates the capability of the proposed method to characterize falling



snowflake features and their temporal dynamics. By using the same methodology as in Sec.4.5 for merging HC-MASC and HC-2DVD classification schemes, we obtain a HSS of 0.80 and an OA of 88.2%, indicating a very good agreement between the two methods.

### 4.5.2   16-17 June 2016

A different scenario is depicted on Fig.14, displaying a mixed-phase precipitation event which occurred over the night of 16th to 17th June 2016. At the beginning, the air temperature was around 4°C and progressively decreased to reach -2°C in the second part of the event. The temperature dropped under the freezing level soon after 22:00 UTC, which corresponds well to the transition between liquid/mixed and solid-phase precipitation detected by both classification methods. In the first part, HC-2DVD identified a few intermittent timesteps of melting snow, which seem to be in good agreement with the proportion

of melting snow evaluated by HC-MASC and displayed on panel **b**). After the transition, the snowfall, mainly composed of aggregates, exhibits short periods of more intense riming around 00:00, 00:45 and between 01:30-02:00 UTC. By merging HC-MASC and HC-2DVD classification schemes, this event obtained a HSS of 0.72 and an OA of 85.6%. Finally, information provided by the ambient temperature measurement and HC-2DVD output made it possible to confidently identify the large proportion of small particles classified by HC-MASC (15:00-22:30 UTC) as rain. As mentioned in section 2.3, MASC fallspeed

measurement unit was not working continuously during the campaign in Davos. This parameter was therefore not included in the classification model. In the future, the inclusion of the particles fallspeed as an additional descriptor in HC-MASC could potentially help to discriminate between raindrops and small solid hydrometeors.

### 5   Conclusions and future perspectives

In this paper, we proposed a novel method to classify individual solid hydrometeor based on MASC particle photographs.

Automatic classification is achieved by means of a cost adjusted penalized MLR model which was trained on more than 3000 images manually labeled by expert users. For this purpose, a large set of geometrical and texture-based descriptors characterizing the snowflakes size, shape and riming extent was developed. A feature selection algorithm was implemented and allowed the identification of the most relevant and discriminating descriptors with respect to the classification task. In order to avoid overfitting and improve the generalization properties of the classifier, the MLR was regularized using an L2 norm as

a prior to the model and we proposed a cost adjustment to increase the performance in presence of imbalanced classes.

Three classifiers were implemented in an independent manner in order to: classify solid-phased hydrometeor type, assess the degree of riming, and detect melting snow. The hydrometeor type was discriminated between six classes adapted from Magono and Lee (1966) and observed in MASC data: columnar crystal, planar crystal, combination of columnar and planar crystals, aggregate, graupel and small particle. The extent of riming was qualitatively evaluated on an ordinal five-level scale which was

in turn used to create a continuous index between zero (none) and one (graupel). Melting snow was detected using a binary classifier between dry and wet snowflakes. The classification performance was evaluated on test data with labels unknown to the classifier and achieved high accuracy, with averaged OA and HSS of 94.7% and 0.93 for the hydrometeor type, respectively.





The binary classification between dry and melting snowflakes performed equally well with an OA of 93.9% and a HSS of 0.86. Discrete classification of the riming degree led to more uncertain results, characterized by an OA and HSS of 75.5% and 0.67, respectively. By using the probabilistic information provided by the MLR model, those values were successfully remapped into a continuous riming index ranging between 0 and 1 with an associated error distribution of mean 0.00 and standard deviation

0.13.

The classification results were further compared and cross-validated with an external and independent method evaluating the dominant type of hydrometeor over timesteps of 5 minutes based on 2DVD data. The two methods were compared over more than 88 hours of collocated measurements and showed a good agreement with an OA and HSS of 72% and 0.56, respectively. Two precipitation events were more thoroughly analyzed and demonstrated the capability of both methods to consistently

identify transitions between liquid, mixed and solid precipitation as well as small-scale variations in the degree of riming of observed snowfall. These events showed a better agreement as indicated by an OA of 88.2% and 85.6% as well as a HSS of 0.80 and 0.72, respectively. Moreover, the proposed method HC-MASC provides additional information which proved to be relevant to gain more insight into the dynamics of snowfall: a continuous riming index as well as the proportions in which each hydrometeor type contributes to the precipitation.

The proposed MLR supervised method is fast (about 15 seconds to process and classify 100 MASC triplets on a recent 4 cores office desktop computer) and repeatable. The image processing, extraction of descriptors and classification procedure was implemented using parallel computing. Once the classifiers are trained, the algorithm can potentially operate in real time. Furthermore, the classification was conducted separately on each photograph of the same particle provided by the MASC. In this way, the method can easily be adapted to any other instrument providing high-resolution images of the particles of interest

(e.g. 2DVD, snow video imager, airborne optical array probes). A feature ranking analysis showed that the hydrometeor type classification could be conducted on binary particle images without significantly altering the classification accuracy. On the other hand, the analysis highlighted that the texture-based descriptors play an essential role in the riming degree evaluation and melting snow detection algorithms.

Finally, the presented method can be used to measure and characterize the microstructural properties of individual falling

snowflakes and paves the way for further microphysical studies based on in-situ measurements. Conditional analysis of particle size distribution, aspect ratio and canting angle as a function of the hydrometeor type could for instance be carried out in order to reduce the uncertainty around their parametrization within numerical weather prediction models. Having a reliable type and riming degree associated with each individual solid particle observed could contribute to refine the relationships between their mass, size and fall velocity, and more generally to improve remote sensing of solid precipitation.

**6   Data availability**

The datasets acquired by the MASC as well as the codes used to process and classify snowflake images can be made publicly available upon request to the authors.





*Acknowledgements.* This work was supported by the Swiss National Foundation (SNF), project number 200021_157210. For the field deployment and maintenance of the instruments in Davos and at Dumont d'Urville (Antarctica), we thank Jacques Grandjean (MeteoSwiss) and Jacopo Grazioli (EPFL-LTE). We are grateful to the French Polar Institute (IPEV) for logistical support during the measurement campaign at Dumont-d'Urville. We also want to thank prof. Satoshi Takahama and prof. Jean-Philippe Thiran from EPFL for sharing their expertise in
5   applied machine learning and for their helpful suggestions.



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




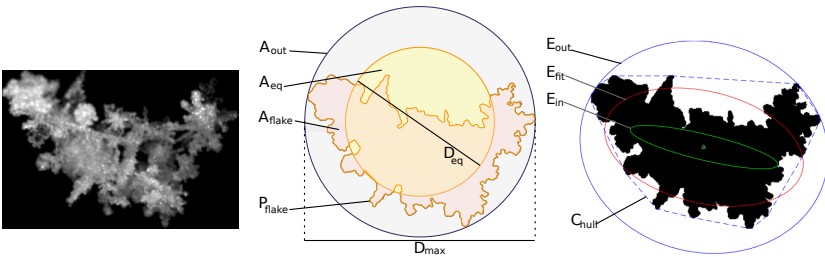

**Figure 1.** Illustration of various particle geometrical descriptors. Left panel: snowflake image obtained with the MASC. Middle panel: particle projected area $A_{\text{flake}}$, perimeter $P_{\text{flake}}$, maximum dimension $D_{\text{max}}$, equivalent-area diameter $D_{\text{eq}}$ and circumscribed circle area $A_{\text{out}}$. Right panel: ellipse fit $E_{\text{fit}}$, inscribed ellipse $E_{\text{in}}$, circumscribed ellipse $E_{\text{out}}$ and convex hull $C_{\text{hull}}$.

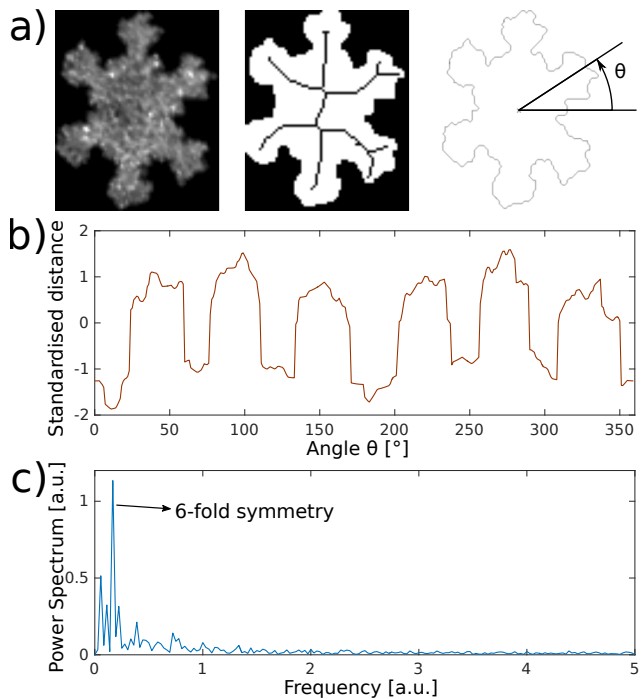

**Figure 2.** Illustration of the particle skeleton and symmetry feature extraction process on a sectored plate: (**a**) original MASC image, particle skeleton and illustration of the angular distance-to-centroid calculation, (**b**) standardised angular distance-to-centroid computed every 1 degree and (**c**) Fourier power spectrum corresponding to the angular signal above. The 7 (including 0) first components of the power spectrum are used as descriptors for hydrometeor classification.



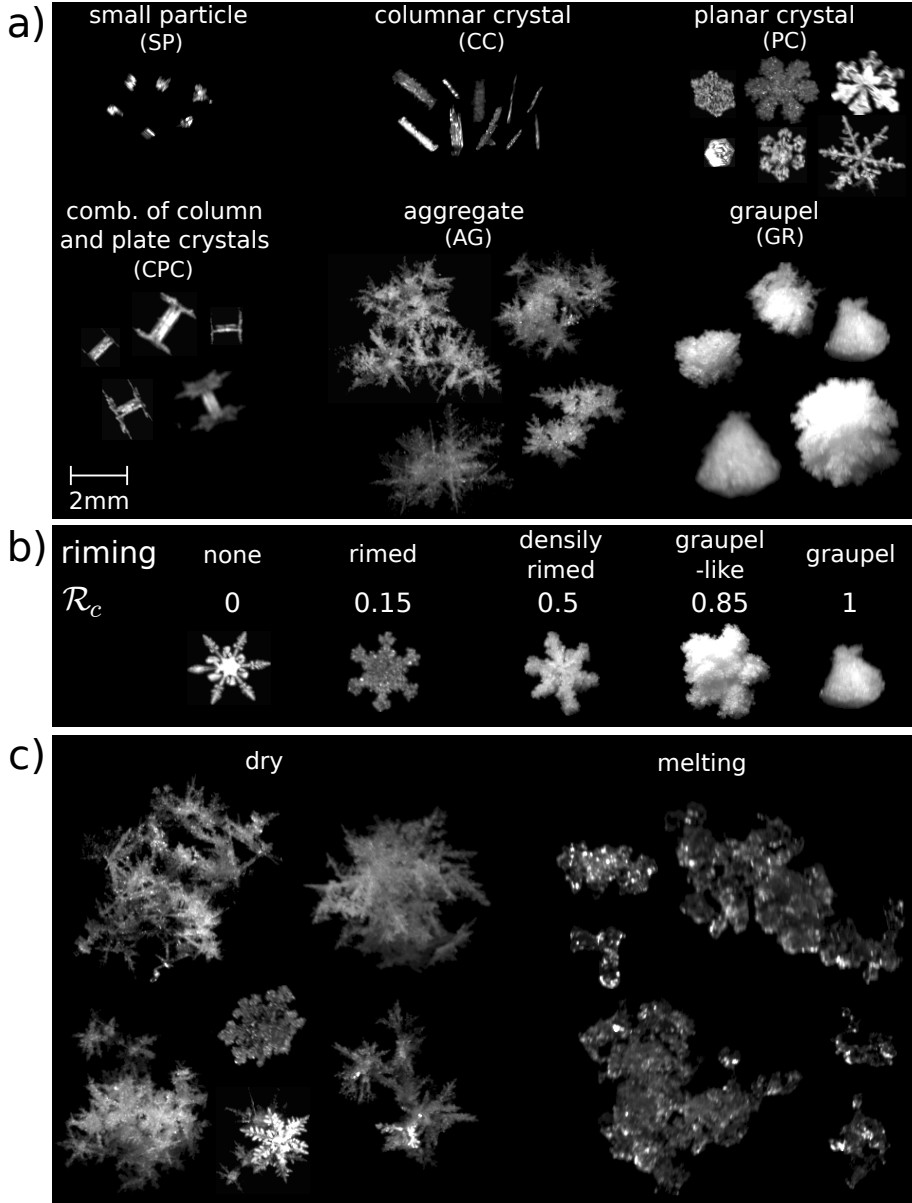

**Figure 3.** Examples of MASC particle images belonging to each hydrometeor class: (**a**) solid hydrometeor type, (**b**) riming scale and (**c**) detection of melting snow.





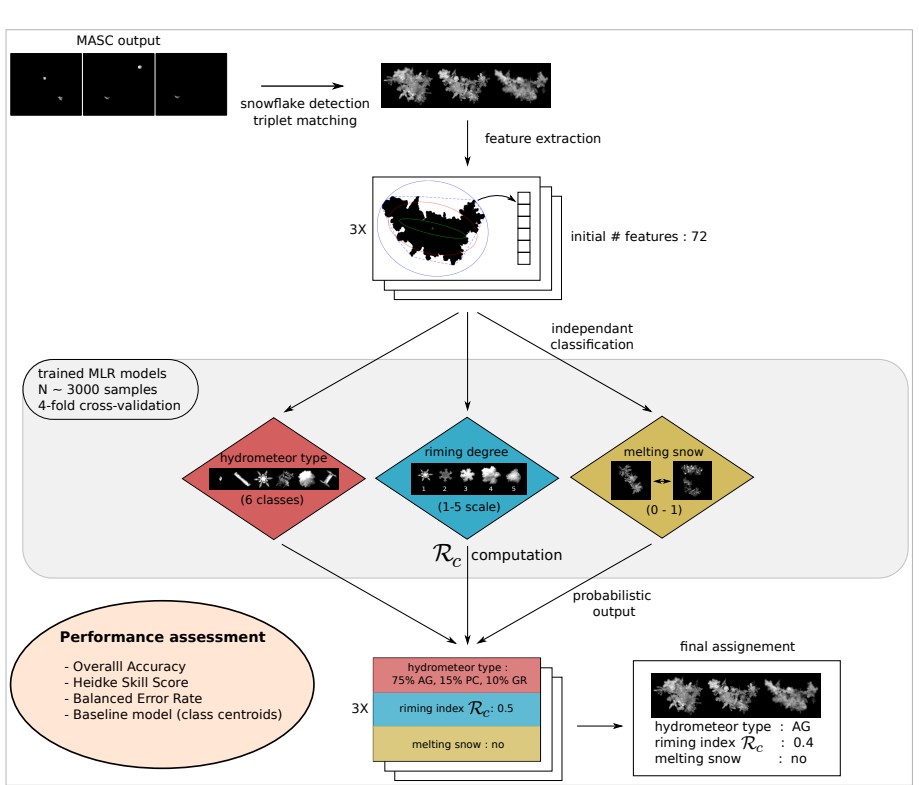

**Figure 4.** Classification procedure for a triplet of images obtained with the MASC.





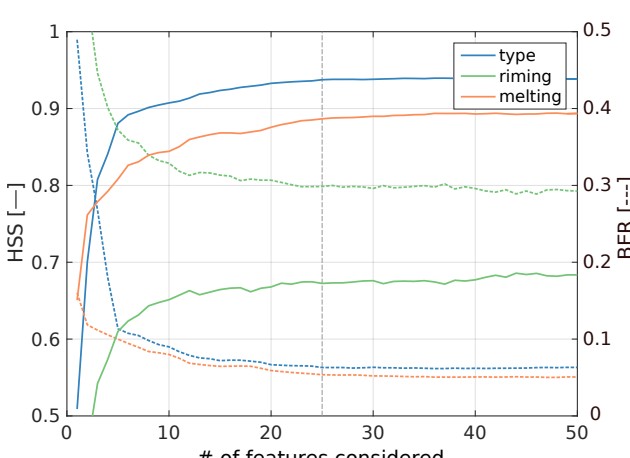

**Figure 5.** Evolution of the HSS (full lines) and BER (dashed lines) as we increase the number of features used in the model. The 3 color codes represent hydrometeor type classification (blue), riming degree (green) and melting snow detection (orange), respectively. Features added on the right of the vertical dashed line are discarded.





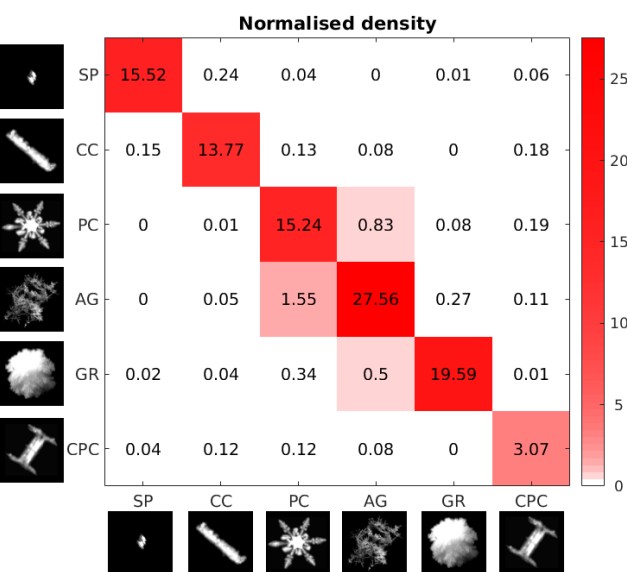

**Figure 6.** Confusion matrix of test data obtained by averaging 10 instances of 4-fold cross-validation. True labels are on the horizontal axis and predictions on the vertical axis. Correct classifications are located on the diagonal. The entries have been normalised so that they sum up to 100.





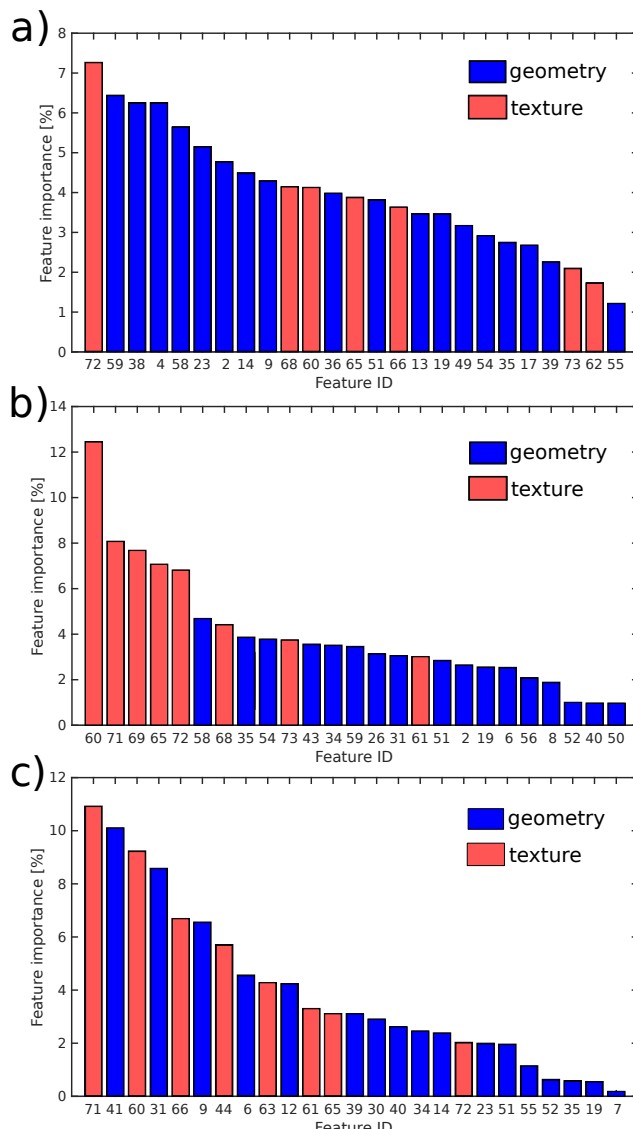

**Figure 7.** Estimation of the importance of each selected feature according to the logistic weights $\beta_d$ for (**a**) hydrometeor type classification, (**b**) riming degree estimation and (**c**) melting snow detection. Descriptors computed only from the binary mask of the particle are displayed in blue (geometry) and descriptors using the textural information are in red (texture). Features ID are referring to Table A1.



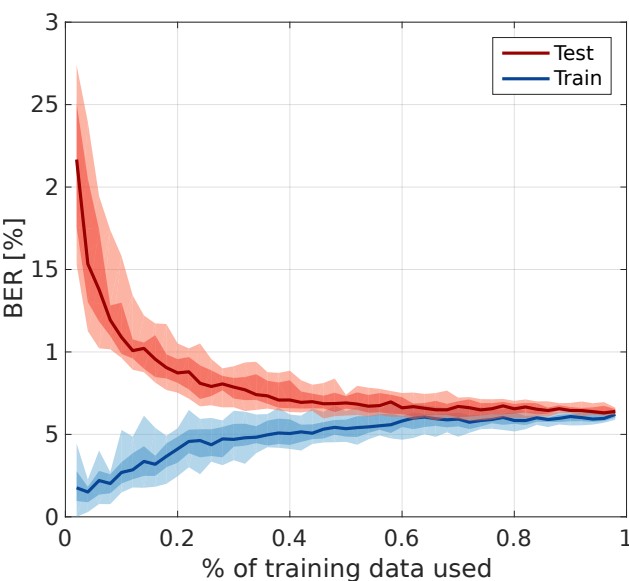

**Figure 8.** Learning curves for the hydrometeor type model, showing the evolution of the train and test BER as a function of the number of training samples used ($100\% = 2429$ images). Dark areas correspond to the 25-75 percentile range computed over 20 iterations of random train-test splitting, light areas extend to the 10-90 percentiles, and bold lines are the medians.



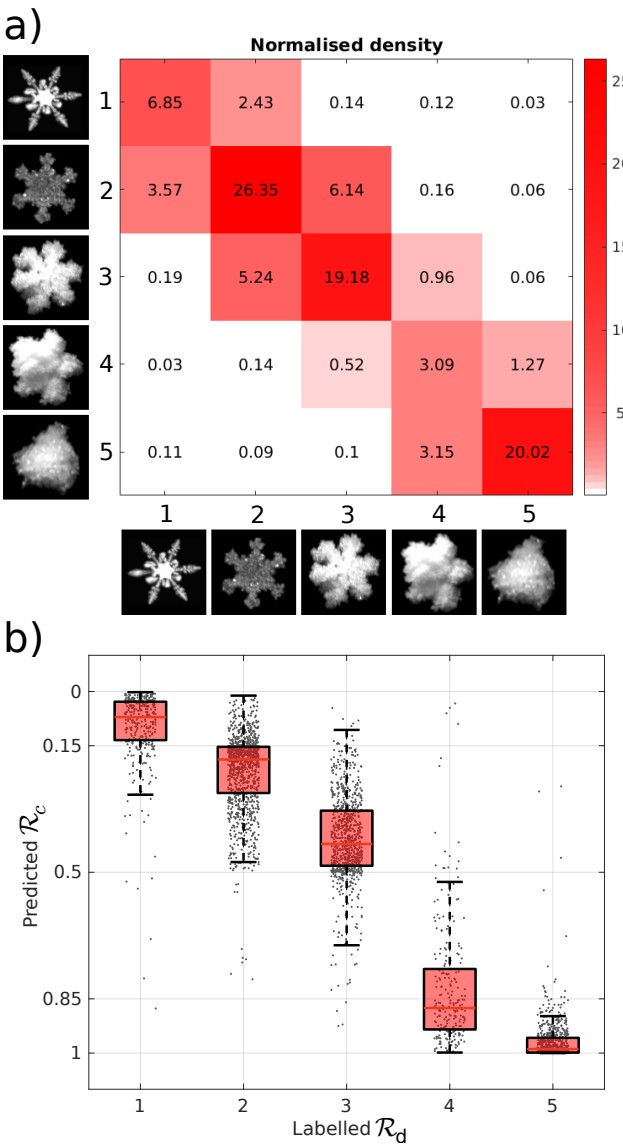

**Figure 9.** Illustration of the riming classification performance. (**a**) Confusion matrix of test data obtained by averaging 10 instances of 4-fold cross-validation. Labelled riming degrees are on the horizontal axis and predicted riming degrees $\mathcal{R}_d$ on the vertical axis. (**b**) Boxplots of predicted riming index $\mathcal{R}_c$ as a function of the labelled riming degree $\mathcal{R}_d$. The gray dots are the test data samples, horizontally spread around each value of $\mathcal{R}_d$ to increase readibility. The red lines are the medians, the boxes correspond to the interquartile range and the whiskers extend to 1.5x the interquartile range.





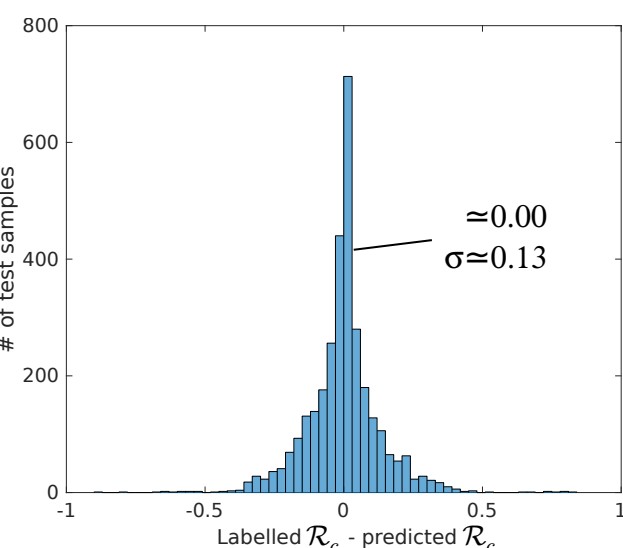

**Figure 10.** Distribution of the classification error between the riming index calculated from the true labels and the riming index predicted by the MLR model. The y-axis shows the number of test samples contained within each bin.

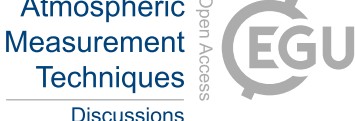

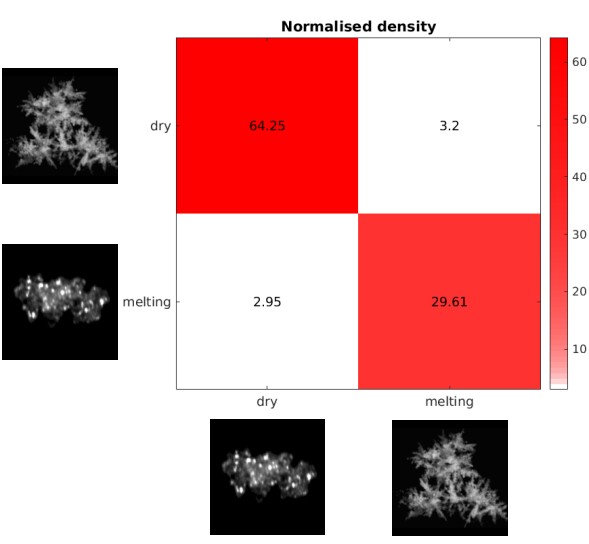

**Figure 11.** Same as in Fig.6 but for melting snow detection.


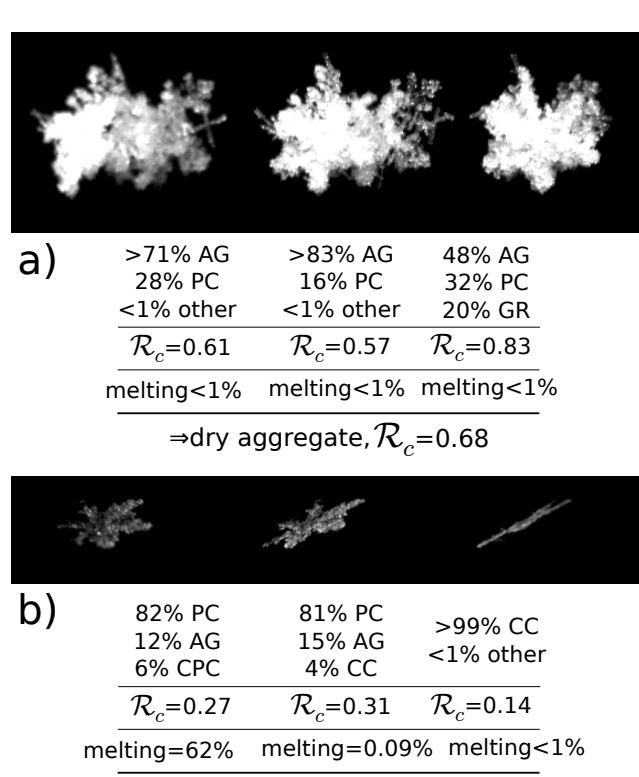

**Figure 12.** Illustration of the prediction merging process applied on 2 unlabeled MASC image triplets. (**a**) A dry, heavily rimed aggregate. (**b**) A dry, moderately rimed planar crystal.




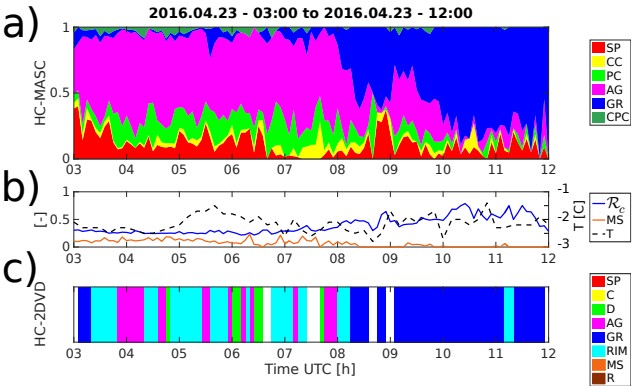

**Figure 13.** Illustration of the MASC and 2DVD hydrometeor classification comparison during a precipitation event recorded on 23 April 2016. Time series of (**a**) proportions of each hydrometeor type as classified with the MLR model applied on MASC data; (**b**) on the left axis: averaged riming index $\mathcal{R}_c \in [0,1]$ and proportion of melting snowflakes $MS \in [0,1]$; on the right axis: ambient temperature $T$, as measured by a weather station located in the DFIR; and (**c**) dominant hydrometeor type recorded by a 2DVD using the method described in (Grazioli et al., 2015). On each panel, data were aggregated on time intervals of 5 minutes.

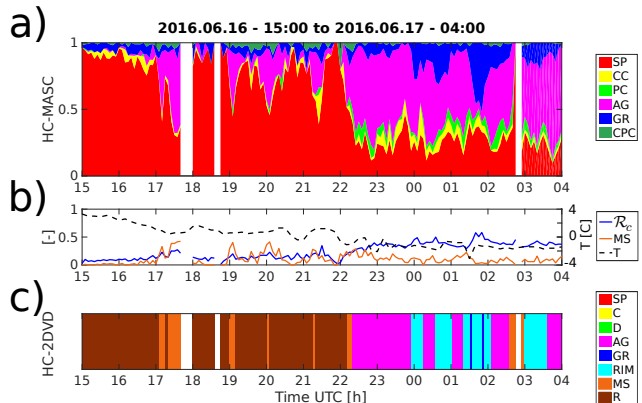

**Figure 14.** Same as on Fig.13 but for the 16-17 June 2016.





**Table 1.** Riming scale and qualitative description of each level. First column is the new riming index introduced in this study and second column its corresponding qualitative riming degree, adapted from Mosimann et al. (1994).

| $\mathcal{R}_c \in [0,1]$ | $\mathcal{R}_d \in [1,5]$ | Coverage of the surface | Description |
|---|---|---|---|
| 0 | 1 (none) | 0% | No cloud droplets on the surface. |
| 0.15 | 2 (rimed) | $\simeq 50\%$ | Up to half of the surface is covered with cloud droplets. |
| 0.5 | 3 (densely rimed) | $\simeq 100\%$ | Cloud droplets are covering the whole surface of the particle but the initial shape is still well conserved. |
| 0.85 | 4 (graupel-like) | $> 100\%$ | Particle initial shape barely recognisable. Surface covered by multiple layers of cloud droplets. |
| 1 | 5 (graupel) | $\gg 100\%$ | Particle initial shape completely transformed into a graupel (lump, conical, or hexagonal). |

**Table 2.** Average accuracy scores obtained for the 3 classification schemes (i.e. hydrometeor type, riming degree and melting snow detection). The numbers indicate the mean $\pm$ the standard deviation calculated over 10 instances of 4-fold cross-validation and applied on test (i.e. unknown) data.

| Method | BER | HSS | OA |
|---|---|---|---|
| **Hydrometeor type** | | | |
| Baseline | $15.2 \pm 1.4\%$ | $0.81 \pm 0.01$ | $84.7 \pm 1.1\%$ |
| MLR | $6.5 \pm 1.1\%$ | $0.93 \pm 0.01$ | $94.7 \pm 0.7\%$ |
| **Riming degree** | | | |
| Baseline | $36.7 \pm 1.9\%$ | $0.47 \pm 0.02$ | $58.7 \pm 1.5\%$ |
| MLR | $30.2 \pm 2.0\%$ | $0.67 \pm 0.02$ | $75.5 \pm 1.5\%$ |
| **Melting snow detection** | | | |
| Baseline | $18.5 \pm 0.8\%$ | $0.56 \pm 0.02$ | $78.9 \pm 1.0\%$ |
| MLR | $7.1 \pm 0.6\%$ | $0.86 \pm 0.01$ | $93.9 \pm 0.6\%$ |





**Table 3.** Confusion matrix obtained by comparing MASC and 2DVD hydrometeor classification during winter 2015-2016 in Davos (CH). Each entry in the matrix corresponds to the dominant hydrometeor type recorded during an interval of 5 minutes measured by both instruments. For HC-2DVD, rain and small particles have been merged (SP+R), as well as rimed particles and aggregates (AG+RIM), respectively.

|  |  | HC-2DVD | | | | | |
|---|---|---|---|---|---|---|---|
|  |  | AG+RIM | D | GR | SP+R | C | MS |
| | AG | **463** | 18 | 97 | 2 | 1 | 19 |
| | PC | 1 | **0** | 0 | 0 | 0 | 0 |
| HC-MASC | GR | 34 | 0 | **51** | 1 | 0 | 3 |
| | SP | 15 | 0 | 28 | **183** | 0 | 24 |
| | CC | 1 | 0 | 1 | 0 | **1** | 0 |
| | MS | 38 | 4 | 1 | 5 | 0 | **66** |





## Appendix A:  Lists of numerical descriptors used for hydrometeor classification

Table A1: Initial list of numerical descriptors provided to the classification model before feature selection. The 7 categories correspond to the different subsections introduced in section 2.3. For each line, the last column indicates if the corresponding feature is retained for (I) hydrometeor type classification, (II) riming degree estimation, (III) melting snow detection after the feature selection as been applied, as detailed in section 3.5.

| Feature ID | Feature description | Category |
|---|---|---|
| | **C1: Particle size and area** | |
| 1 | Particle projected area | - |
| 2 | Particle perimeter | I,II |
| 3 | Particle mean dimension | - |
| 4 | Particle maximum dimension | I |
| 5 | Particle equivalent-area diameter | - |
| 6 | Particle porous area (accounting for holes) | II,III |
| 7 | Porous area over total area ratio | III |
| | **C2: Elliptical approximations** | |
| 8 | Fitted ellipse major axis | II |
| 9 | Fitted ellipse minor axis | I,III |
| 10 | Fitted ellipse area | - |
| 11 | Fitted ellipse orientation | - |
| 12 | Fitted ellipse aspect ratio | III |
| 13 | Fitted ellipse eccentricity | I |
| 14 | Particle compactness (projected area to fitted ellipse area ratio) | I,III |
| 15 | Inscribed ellipse major axis | - |
| 16 | Inscribed ellipse minor axis | - |
| 17 | Inscribed ellipse area | I |
| 18 | Circumscribed ellipse major axis | - |
| 19 | Circumscribed ellipse minor axis | I,II,III |
| 20 | Circumscribed ellipse area | - |
| 21 | Inscribed/fitted ellipse major axis ratio | - |
| 22 | Inscribed/fitted ellipse minor axis ratio | - |
| 23 | Inscribed/fitted ellipse area ratio | I,III |
| 24 | Inscribed/circumscribed ellipse major axis ratio | - |
| 25 | Inscribed/circumscribed ellipse minor axis ratio | - |
| 26 | Inscribed/circumscribed ellipse area ratio | II |
| 27 | Fitted/circumscribed ellipse major axis ratio | - |



| 28 | Fitted/circumscribed ellipse minor axis ratio | - |
|----|------------------------------------------------|-----|
| 29 | Fitted/circumscribed ellipse area ratio | - |
| | **C3: Particle shape** | |
| 30 | Particle roundness (area to circumscribed circle area ratio) | III |
| 31 | Circumscribed circle perimeter to particle perimeter ratio | II,III |
| 32 | Particle rectangularity (area to bounding box area ratio) | - |
| 33 | Bounding box width | - |
| 34 | Bounding box height | II,III |
| 35 | Bounding box perimeter to particle perimeter ratio | I,II,III |
| 36 | Bounding box aspect ratio | I |
| 37 | Bounding box eccentricity | - |
| 38 | Particle solidity (area to convex hull area ratio) | I |
| 39 | Particle convexity (convex hull perimeter to particle perimeter ratio) | I,III |
| 40 | Number of vertex in the convex hull | II,III |
| 41 | Particle perimeter to equivalent-area circle perimeter | III |
| 42 | Particle fractal dimension (boxcounting method) | - |
| 43 | Particle fractal index (as in Grazioli et al., 2014) | II |
| | **C4: Morphological skeleton** | |
| 44 | Skeleton number of ending points | III |
| 45 | Skeleton number of branching points | - |
| 46 | Skeleton length to particle perimeter ratio | - |
| 47 | Skeleton length to particle area ratio | - |
| | **C5: Rotational symmetry** | |
| 48 | Standardized distance to centroid Fourier power spectrum comp. P0 | - |
| 49 | Standardized distance to centroid Fourier power spectrum comp. P1 | I |
| 50 | Standardized distance to centroid Fourier power spectrum comp. P2 | II |
| 51 | Standardized distance to centroid Fourier power spectrum comp. P3 | I,II,III |
| 52 | Standardized distance to centroid Fourier power spectrum comp. P4 | II,III |
| 53 | Standardized distance to centroid Fourier power spectrum comp. P5 | - |
| 54 | Standardized distance to centroid Fourier power spectrum comp. P6 | I,II |
| 55 | #max(P0 to P6) | I,III |
| 56 | Ratio between P6 and max(P0 to P6) | II |
| 57 | Distance to centroid mean | - |
| 58 | Distance to centroid standard deviation | I,II |
| 59 | Distance to centroid standard deviation over mean ratio | I,II |
| | **C6: Texture operators** | |
| 60 | Particle mean pixel brightness | I,II,III |
| 61 | Particle maximum pixel brightness | II,III |





| | | |
|---|---|---|
| 62 | Image contrast | I |
| 63 | Particle pixel brightness standard deviation | III |
| 64 | Brightness histogram entropy | - |
| 65 | Average gray-level local variance (3x3 moving window) | I,II,III |
| 66 | Average gray-level local range intensity (3x3 moving window) | I,III |
| 67 | Energy of Laplacian (3x3 moving window) (Pertuz et al., 2013) | - |
| 68 | Sum of wavelet coefficients (Pertuz et al., 2013) | I,II |
| 69 | Particle complexity (Garrett and Yuter, 2014) | II |
| | **C7: Co-occurrence matrix** | |
| 70 | Haralick angular second moment | - |
| 71 | Haralick contrast | II,III |
| 72 | Haralick correlation | I,II,III |
| 73 | Haralick homogeneity | I,II |