# Peer review of "Solid hydrometeor classification and riming degree estimation from pictures collected with a Multi-Angle Snowflake Camera"

_Atmospheric Measurement Techniques, 2016_

## Referee Comment (RC1) · Anonymous Referee #1 · 19 Jan 2017

Review of "Solid hydrometeor classification and riming degree estimation from pictures collected with a Multi-Angle Snowflake Camera", by Prez, Roulet, and Berne, submitted to Journal of Atmospheric Measurement Technology.

This is a very well thought out study and the article is well written. It describes a sophisticated set of criteria, based on machine learning, with the algorithms developed from Principle Component Analysis and Linear Discriminant Analysis, to classify the shapes of ice crystals imaged by the MASC probe into one of six classes of the categories developed by Magono and Lee. My comments are fairly minor, but should be considered by the authors, and as such I recommend that the article is accepted for publication with minor revision.

[Figure]

Major Comments

1. Korolev (GRL, 1999) states that "Using new technology imaging instrumentation with a resolution of 3 microns, recent observations in Arctic clouds have shown that such pristine habits only describe approximate 3% of the particles. The measurements were made from an aircraft during April 1998 and over a temperature range of 0 to -450C". In view of this finding, does it make sense to classify particles into habit categories?

2. You may want to cite the article by Durore (1982) in J. Rec. Atmos. that uses a Fourier analysis technique to classify the habits of article particles, and briefly discuss the merits of that approach and why it was not used in your study.

3. It would have been very useful to also describe the particles by their fractal dimensions as this dimension can be used to link the particles' cross-sectional area to mass.

4. I feel strongly that, given that the MASC also measures the particle fall velocity, it would be very useful to rebuild your algorithms with that property also considered.

Minor Comments

1. Page 1, line 1: Remove "a" 2. 1, 18. "interpretation of radar retrievals" to "development of radar retrieval algorithms". 3. 1, 18. "estimation" to "retrieval". 4. 4, 4. "recall" to "cite".

---

## Referee Comment (RC2) · Anonymous Referee #2 · 27 Jan 2017

In their study, the authors present a new method to classify snowflakes based on MASC observations. The method is very valuable and well presented. I appreciate the high quality of the paper, I have never reviewed a paper with only one slip of the pen before.

I have only few comments, most of them concern parts where I had trouble understanding what was exactly done. Note that I'm not familiar with multilinear logistic regression, therefore I cannot evaluate whether every step of the methodology is 100% correct.

I recommend the paper to be published after minor revision.

p3, l6: hyperspectral images of what? Sec 3.5 Did I understand that correctly that the descriptors are ordered here by information content? If yes, please say so clearly.

This also applies to p13 l5 "for each subset of features" which sounds like you would have chosen a set of feature randomly. Maybe the authors can say something like "Subsets of features were created by subsequently adding features from highest to lowest information content as presented in sec 3.5. For each subset…"

p12, l1: Please define 4-fold cross-validation

p14, l7-15: I don't understand how you can estimate BER for the training dataset? Isn't BER supposed to be perfect for the training dataset?

p14, l1: The only slip I found: on -> in

p16, l9: Other instruments (e.g. PIP) have only one camera. Can you quantify the benefit of using three instead of one camera? Similar how you quantified the importance of gray scale images?

Fig 2: Please define a.u. And can you draw a vertical line after 7 components in c)?

Fig 7: I would recommend to add titles to the subfigures: a) type classification, b) riming etc.

Fig 8: x-axis is not in %.

---

## Referee Comment (RC3) · Anonymous Referee #3 · 12 Feb 2017

Overall a well-conceived and well-written paper that uses machine learning to classify images from a Multi-Angle Snowflake Camera (MASC). Three sets of classification categories are used: 5 hydrometeor types, degree of riming, and whether a particle is melted or not.

A few areas need additional citations:

1) Page 2, Suggest adding reference to previous work on habit classification for optical probe data that used area ratio, the area of the particle divided by the area of the circle that circumscribes the maximum dimension of the particle. Relevant citations are Heymsfield and Parrish 1979; Heymsfield and Kajikawa 1987; Heymsfield and McFarquhar 1996; Heymsfield et al. 2002

2) Page 4, There is another relevant Nurzy'nska et al. paper on shape parameters that needs to be cited, Meteorol. Appl. 20: 257–265 (2013).

Clarifications:

Given the limited depth of field of the MASC cameras, many particles in MASC images are out of focus. This has ramifications for any automated image processing algorithm.

3) Please clarify how in focus (non-blurry) images were determined for both the training set and for the unlabeled MASC data. Do you use some objectively defined in-focus parameter? If so please define it and provide information on the distribution of this parameter for the training set and the unlabeled MASC data.

4) Please clarify what are the specific requirements for input images. Please include how sharp (in focus) do the MASC particle images need to be for the method to be reliable.

5) Please clarify, is the intention and/or requirement that this method is applied to triplets of in-focus images.

6) Regarding the training set (Section 3.2): What criteria were used to select images for the training set. Over how many storms and what range of environmental conditions.

Section 4.5 Comparison with existing classification method. The differences between the instrument output of the MASC (33 um spatial resolution gray scale image) and 2DVD (0.2 mm spatial resolution binary image) as well as large difference in sample size for each 5 min interval (at least 30 non-blurry MASC images to 300 2DVD particles) makes direct comparison problematic. Very close correspondence of classifications is not expected.

7) The small particle classification into ice or rain relies on a temperature measurement. Should temperature be added to the characteristics in the machine learning.

8) Another possible reason for differences is that often ice particles of different classifications reach the surface simultaneously (Stark et al. 2013, MWR). It is possible that 30 MASC classified particles may not be representative of the 300 particles observed by the 2DVD.

9) Rather than showing only the one predominant classification for each 5 min period from 2DVD can the distribution of 1-min classifications be shown. This would provide a more direct comparison to distribution of classifications from MASC images.

10) Page 18, for "temporal dynamics" do you mean "temporal variability"?

I am less concerned than other reviewer about not using fall speed as part of classification. When multiple particles are in one image, the one MASC measured fall speed cannot be clearly attributed to any one or set of those particles. Hence even if future versions of the algorithm use fall speed, it will not be available for individual particles cropped from MASC images with multiple particles.

---

## Author Comment (AC3) · 22 Mar 2017

Please see attached file.

Please also note the supplement to this comment:
http://www.atmos-meas-tech-discuss.net/amt-2016-417/amt-2016-417-AC3-supplement.pdf

---

## Author Comment (AC4) · 22 Mar 2017

See attached file.

Please also note the supplement to this comment:
http://www.atmos-meas-tech-discuss.net/amt-2016-417/amt-2016-417-AC4-supplement.pdf

---

## Author Response (AR1)

[revised manuscript text omitted]

This article introduces a new method which makes use of MLR to automatically classify individual hydrometeors observed by a MASC (and potentially other imaging sensors), based on a large set of geometrical and texture-based features developed for this purpose. The paper is structured as follows: section 2 describes the experimental setup and the MASC image processing procedure. Section 3 presents the proposed classification model. The main results, a comparison with independent measurements as well as some classification examples are given in section 4. Finally, a conclusion summarising the work and presenting some future perspectives is drawn in section 5.

**2 Data description**

**2.1 Data collection**

Images used to develop and evaluate the hydrometeor classification were obtained with a MASC. The MASC data were collected during two different measurement campaigns organized during the winter 2015-2016. The first campaign took place from October 2015 to June 2016 in Davos, Switzerland. During that time, the MASC was deployed in a Double Fence Intercomparison Reference (DFIR) at a meteorological test site located at 2540 m a.s.l. Also present in the DFIR during the measurement campaign were a 2DVD and a weather station. The second campaign took place from November 2015 until January 2016 on the Antarctic French base of Dumont d'Urville, in the framework of the Antarctic Precipitation, Remote Sensing from Surface and Space project (APRES3, http://apres3.osug.fr). Collocated measurements from a weighing precipitation gage and a weather station were also collected. In total, more than two million of MASC images were collected and processed during these two measurement campaigns.

**2.2 MASC instrument**

The MASC is a ground-based instrument which automatically takes high resolution and stereoscopic photographs of hydrometeors in free fall while measuring their fall velocity. Its working mechanism being extensively explained in Garrett et al. (2012), we will only  mention the main aspects here. The imaging unit is composed of three high resolution cameras attached to a ring structure and separated by an angle of 36°. Each camera points at an identical focal point lying in the middle of the ring structure, at approximatively 10 cm from the cameras. The triggering unit is composed of two pairs of horizontally aligned near-infrared emitter-receiver arrays, delimiting a measuring cross-section of approximatively 2.5 cm$^2$. Particles falling successively through both arrays are detected and trigger the three cameras as well as three spotlights used to illuminate the target. The two MASCs used in the present study were using identical 2448x2048 pixels cameras mounted with 12.5 mm lenses. The cameras' aperture and exposure time were adjusted in order to maximize the contrast on hydrometeor photographs while preventing motion blur effects. With these settings, in-focus image resolution was measured to be about 33 $\mu$m per pixel using a graduated calibration target.

As the instrument triggering area is larger than the cameras depth of field, many MASC images are out of focus and appear blurred. This is important as the blurriness in the images will influence the value of certain descriptors introduced in Sec. 2.3 and in turn, the classification performance. Due to the large variety of shape and structure in the observed hydrometeors, it is however difficult to automatically distinguish out of focus images. For this study, efforts were made to establish a dimensionless empirical quality index $\xi$ based on particle size, brightness and interpixel variability which quantifies the blurriness present in the image. Typical values for $\xi$ lie within 7 (blurry) and 12 (sharp). The exact methodology used to develop this index as well as some illustrations are detailed in appendix B.

**2.3 Image processing and feature extraction**

Similar to the human brain, a computer algorithm requires a set of criteria to rely upon for image classification. In the present case, this set of criteria takes the form of numerical descriptors, commonly called features in machine learning studies and computed from the particle photographs. Regardless of the classification method used, extracting an exhaustive and relevant set of features and avoiding redundancy are two essential steps as they will strongly affect the performance of the classifier. Because it is *a priori* impossible to know exactly what features are relevant to the target concept (i.e. hydrometeor classification), a large set of 72 descriptors derived from the particle size, shape, and textural information was introduced. Several of them have already been used for hydrometeor identification purposes in previous works (e.g., Lindqvist et al., 2012; Nurzyńska et al., 2012, 2013; Grazioli et al., 2014; Schmitt and Heymsfield, 2014). As we experienced some issues with the MASC fallspeed measuring unit during the campaign in Davos, this parameter was discarded in the proposed methodology. Additionally, keeping the classification independent from this variable makes it possible to study *a posteriori* the relationship between hydrometeor type, geometry and fallspeed in an objective manner.

[revised manuscript text omitted]

5      Melting falling snow is characterized by eroded particle outlines as well as the presence of liquid water droplets forming on their surface. On MASC images, these liquid water droplets appear as glints of reflection identified by their small size and saturated pixel values. It is therefore possible to detect melting snowflakes using the geometrical and textural descriptors detailed in section 2.3. The detection is achieved through a binary classification between *dry* and *melting* hydrometeors as illustrated on Fig.3**c**. Because the reflection in liquid water is small and practically independent from the size of the drop, it is

10   very difficult to differentiate raindrops from small particles on the basis of the image only. As a result, a liquid precipitation event will be identified by a proportion of small particles close to 100%.

**3.2   Training set**

The preparation and labeling of a training set is a very important step as it strongly influences the learning phase and directly impacts the ability of the model to generalize to unknown data. For the present work, a dataset of $N_{\text{label}}$ =3712 MASC

15   particle images was selected in an effort to reflect the proportions between the hydrometeor classes, as observed during the campaigns. In order to cover a large range of environmental conditions, hydrometeors included in the training set were selected from more than 40 snowfalls recorded in the Swiss Alps and in Antarctica. In terms of image quality, only particles character-ized by a quality index $\xi \geq 9$ have been included in the training set. This threshold corresponds to a proportion of 50% of the whole dataset collected in Davos during the winter 2015-2016, as illustrated in appendix B.

[revised manuscript text omitted]

**Appendix B: Image quality index $\xi$**

As mentioned in section 2.2, the MASC is subject to out of focus and blurring issues. On average, more than 50% of the snowflakes imaged are out of focus and appear blurred. Basic geometrical descriptors like particle maximum dimension, axis ratio and orientation are not much affected by blur issues. However, they can significantly modify the values of texture-based features and become a limiting factor for demanding tasks like hydrometeor classification and riming degree estimation. It is therefore necessary to develop tools to quantify the blurriness present in MASC images in order to be able to filter out lower quality images depending on the application.

This matter has been addressed through the introduction of an empirical in-focus parameter called quality index $\xi$. For that purpose, a selection of 500 MASC snowflake images have been manually classified into five categories, from very blurry to very sharp. The quality index was built upon the following observations: out of focus particles tend to have lower brightness, weaker contrast, simpler outline and lower internal variability. The final form of the quality index, established from a trial and error procedure by trying to best discriminate the five categories of blurriness introduced above, is given by:

$$\xi = \log \left( D_{\text{mean}} \cdot \frac{P}{2\pi r_{\text{eq}}} \cdot \frac{\mathcal{L} + \tilde{\mathcal{L}}}{2} \cdot \frac{<\sigma> + <\tilde{\sigma}>}{2} \right),$$

where $D_{\text{mean}}$ denotes the particle mean diameter (feature #3 in Table A1), $\frac{P}{2\pi r_{\text{eq}}}$ the ratio between the particle's perimeter and the perimeter of an equivalent area circle, $\mathcal{L}$ the energy of Laplacian (feature #67) and $<\sigma>$ the average gray-level local standard deviation (feature #65). The $\tilde{\mathcal{L}}$ and $<\tilde{\sigma}>$ indicate that the operators were calculated after image enhancement using a contrast-limited adaptive histogram equalization method (CLAHE, Zuiderveld 1994). In this way, a higher quality index is assigned to dark but sharp snowflakes (i.e. easily identified by visual inspection). The logarithm is applied in order to rescale the index on a more practical range of values. Figure 15 illustrates values of $\xi$ for the 500 snowflakes manually classified

and shows the capability of the index to distinguish between the different categories (with some overlap). Figure 16 displays the distribution of $\xi$ for the training set used for classification as well as for the whole measurement campaign in Davos. As mentioned in Sec. 3.2, only images with a quality index $\xi \geq 9$ have been included in the training set, which corresponds to a 50% proportion of the whole dataset collected in Davos during the winter 2015-16. If we move the threshold to $\xi = 9.5$ (where

5   the two density functions intersect on Fig.16), this proportion drops to 30%.

[Figure]

**Figure 15.** Distribution of the quality index $\xi$ within the 5 categories of snowflakes manually classified.

[Figure]

**Figure 16.** Empirical probability density function of the quality index $\xi$ for the training set and for the whole measurement campaign in Davos. The threshold at $\xi = 9$ is highlighted.

**Solid hydrometeor classification and riming degree estimation from pictures collected with a Multi-Angle Snowflake Camera**

**amt-2016-417**

Christophe Praz, Yves-Alain Roulet, and Alexis Berne

March 22, 2017

**Responses to reviewers**

First, we would like to thank the editor and the reviewers for their constructive comments. In the present document, we provide our responses to the comments of the three reviewers. The comments of the reviewers are reported in *italic*, our responses in normal font and the corresponding modifications in the manuscript in blue. A few additional (but minor) changes independent from the reviewers' comments have also been made on the manuscript. These modifications are:

- Figure 11: on the x-axis, the illustrations for dry and melting snow were inverted. This is now corrected.

- In section 3.1 p.8 l.3, we added a sentence to clarify that the riming degree is undefined in case of small particles: Note that for hydrometeors identified as small particles (i.e. hydrometeors which are too small to be resolved by the MASC), the riming degree estimation is unreliable and therefore discarded.

- Feature #65 was renamed Average gray-level local standard deviation in Table A1.

- In section 3.2 p.8 l.24: 156 images have been added to the training set for the riming degree estimation in an effort to better represent graupel-like particles ($\mathcal{R}_d = 4$). As a result, the classification performance are slightly different (by less than 1%). Score values in Table 2 and in the text (abstract, Section 3.2, Section 4.2, Section 5) as well as Figure 9 and 10 have been updated accordingly. These are very minor modifications that do not change the interpretation of the results.

**Anonymous referee #1**

*This is a very well thought out study and the article is well written. It describes a sophisticated set of criteria, based on machine learning, with the algorithms developed from Principle Component Analysis and Linear Discriminant Analysis, to classify the shapes of ice crystals imaged by the MASC probe into one of six classes of the categories developed by Magono and Lee. My comments*

*are fairly minor, but should be considered by the authors, and as such I recommend that the article is accepted for publication with minor revision.*

We thank the reviewer for their positive comment. Even though we refer to some studies using Principle Component Analysis and Linear Discriminant Analysis in the manuscript (e.g. Lindqvist et al. (2012)), please note that the methods used in the present work are Multinomial Logistic Regression for the classification part and greedy forward selection for the feature selection part.

**Major comments**

*1. Korolev (GRL, 1999) states that "Using new technology imaging instrumentation with a resolution of 3 microns, recent observations in Arctic clouds have shown that such pristine habits only describe approximate 3% of the particles. The measurements were made from an aircraft during April 1998 and over a temperature range of 0 to -450C". In view of this finding, does it make sense to classify particles into habit categories?*

We also observed a clear majority of aggregated and graupel forms in our MASC data. However, statistics performed over the whole measurement campaigns in Davos (Swiss Alps) and Dumont d'Urville (Antarctica) revealed a higher proportion of pristine habits compared to Korolev et al. (1999): 15% with a prevalence of planar crystals for Davos and 16% with a prevalence of columnar crystals for DDU, respectively. We do not have a complete explanation of these discrepancies but the different latitude, conditions of observation (ground based images of precipitating snow instead of cirrus and stratiform cloud particles) and duration of the measurement period (14 flights within a month for Korolev, more than 40 snowfall events in this study) certainly play a role. Parts of these findings (for Antarctica) are presented in another article currently under review in *The Cryosphere* (Grazioli et al., 2017). The fact that the method achieves very high classification accuracy (see Figure 6 in the manuscript) when applied to pristine habits is an additional argument for keeping these classes in the scheme.

*2. You may want to cite the article by Duroure (1982) in J. Rec. Atmos. that uses a Fourier analysis technique to classify the habits of article particles, and briefly discuss the merits of that approach and why it was not used in your study.*

We thank the reviewer for pointing out the work from Duroure (1982) that we were not aware of. However, we did not manage to get access to the original article online. From what is written about the technique in Korolev et al. (2000) : *Duroure (1982) performed a radial analysis of particle image outlines on a selected dataset. A Fourier transform was then used to extract shape discrimination from the image parameters of the analyzed dataset*, it seems that the technique is very similar to the rotational symmetry descriptors introduced in the present work (category C5, detailed on p.6 l.4-11). As we could not find the original article, we however decided to not refer to it.

*3. It would have been very useful to also describe the particles by their fractal dimensions as this dimension can be used to link the particles' cross-sectional area to mass.*

The hydrometeors fractal dimension is already estimated and used as a descriptor in our method in two ways. First, the fractal dimension is calculated using a boxcounting method (Sarkar and Chaudhuri, 1994) and then, it is also estimated by a more theoretical index $FI = 2\frac{\log P/4}{\log A}$, where

$P$ and $A$ are the particle perimeter and area, respectively (Grazioli et al., 2014). These descriptors are introduced and referred to on p.5 l.24-26 and listed in Appendix A (Feature ID 42 and 43). To give the reviewer an idea, typical values for MASC particle fractal dimension computed with the boxcounting method are lying between 1.5 (dendritic crystals, complex aggregates) and 2 (fully developed graupels).

*4. I feel strongly that, given that the MASC also measures the particle fall velocity, it would be very useful to rebuild your algorithms with that property also considered.*

As pointed out by Reviewer #3 and detailed by Garrett et al. (2012), MASC measured fall speed cannot be attributed unambiguously for images containing more than one particle, which happens quite frequently. As a result, if the classifier takes into account the measured fall speed, these particles would have to be either discarded or processed with a different classifier which does not consider fall speed. In our opinion, both alternatives are unsatisfactory. Moreover, Garrett and Yuter (2014) demonstrated that the fall speeds measured with the MASC were strongly influenced by local conditions such as wind, temperature and turbulence. As a result, we do not expect to see a clear relationship between the hydrometeor type (or the degree of riming) and the observed fall speed (which can be very different from the theoretical terminal velocity) at the scale of the individual particle. Moreover, including particles fall speed into the classification method would make it difficult to apply a classifier trained on one field campaign under specific conditions onto data collected at a different place with a different setup and potentially under very different conditions (altitude, temperature, wind, local turbulence, . . . ). As the labeling and classification training steps are long and time consuming, it is not realistic to train a new classifier for each measurement campaign.

We nonetheless do agree that the fall speed measurement provided by the MASC is insightful and should be considered when interpreting the output of the classification. As this variable is not utilized in the classification process, it allows us to objectively study the relationship between particle type and fall velocity (it would not be the case if the particle type would depend on its fall velocity). To clarify this point, a sentence has been added in Section 2.3 p.4 l.29:

Additionally, keeping the classification independent from this variable makes it possible to study a posteriori the relationship between hydrometeor type, geometry and fall speed in an objective manner.

**Minor comments**

*1. Page 1, line 1: Remove "a" 2. 1, 18. "interpretation of radar retrievals" to "development of radar retrieval algorithms". 3. 1, 18. "estimation" to "retrieval". 4. 4, 4. "recall" to "cite".*

The suggested modifications have been implemented into the manuscript (except 3. to avoid the repetition of the word "retrieval" in the same sentence).

**Anonymous referee #2**

*In their study, the authors present a new method to classify snowflakes based on MASC observa-*

*tions. The method is very valuable and well presented. I appreciate the high quality of the paper, I have never reviewed a paper with only one slip of the pen before. I recommend the paper to be published after minor revision.*

We thank the reviewer for their positive comment.

*I have only few comments, most of them concern parts where I had trouble understanding what was exactly done. Note that Im not familiar with multilinear logistic regression, therefore I cannot evaluate whether every step of the methodology is 100% correct.*

*p3, l6: hyperspectral images of what?*

Hyperspectral images of land properties. The sentence has been rephrased as:

For instance, it has been applied recently for land-use land-cover classification based on airborne hyperspectral images of land properties (Li et al., 2010) and ancillary soil data (Kempen et al., 2009).

*Sec 3.5 Did I understand that correctly that the descriptors are ordered here by information content? If yes, please say so clearly.*

Yes, this is the general idea, assuming that the Heidke Skill Score is an adapted tool to measure the information brought by the descriptors. A sentence has been added on p.11 l.16 to clarify this idea:

In other words, the method sorts the descriptors by information content.

*This also applies to p13 l5 "for each subset of features" which sounds like you would have chosen a set of feature randomly. Maybe the authors can say something like "Subsets of features were created by subsequently adding features from highest to lowest information content as presented in sec 3.5. For each subset..."*

We thank the reviewer for their suggestion. As proposed, we rephrased the sentence on p.13 l.10 as:

Subsets of features were created by subsequently adding descriptors from highest to lowest information content as presented in Sec.3.5 (greedy forward feature selection).

*p12, l1: Please define 4-fold cross-validation*

We added the following sentences on p.12 l.8:

$K$-fold cross-validation is a validation method commonly used in machine learning for assessing the generalization capability of a predictive model. It consists in partitioning a labeled dataset into $K$ even complementary subsets and subsequently use each of them as a test set while keeping the $K - 1$ others as a training set.

*p14, l7-15: I dont understand how you can estimate BER for the training dataset? Isnt BER supposed to be perfect for the training dataset?*

In an ideal case where the features introduced perfectly describe and discriminate the training dataset between the different classes present in the model, yes. In practice, it is almost never the case. As mentioned multiple times in the manuscript, Multinomial Logistic Regression (MLR) is a linear model, in the sense that it works directly in the original feature space. In our case, we decided to keep 25 features for the classification task. It means that the MLR model will try to find the best linear boundary (hyperplane of dimension 24) which discriminates the training dataset in the 25 dimensions feature space (based on a maximum likelihood estimator). If the training dataset is not linearly separable in the feature space, the training BER will be different from 0. Of course, we can always complicate the model and/or increase arbitrary the number of input features in order to decrease the training BER, but this is not desired because it exposes us to the curse of dimensionality and the risk of overfitting. Think about the following example: you have a time series $y(t)$ which follows a quadratic relationship with some noise. You want to fit a predictive model to this time series. If you fit a second order polynomial equation to the data points, you will get an error which is non-zero (because of the noise). If you fit a 9-order polynomial equation, it is likely that you will be able to perfectly fit your datapoints (i.e. your training set). However, the predictive capability of this complex model will be extremely bad, because we fitted the noise.

The fact that we observe a convergence of the training and test BER on Figure 8 is very encouraging. It means that the predictive capability of the model on the test set is as good as possible. The "residual" BER ($\sim 6.5\%$) corresponds to the part of the data which cannot be explained by the features used in the model, or by the model itself (MLR).

*p14, l1: The only slip I found: on $\rightarrow$ in*

We could not find the mistake. There was no "on" on p.14, l.1.

*p16, l9: Other instruments (e.g. PIP) have only one camera. Can you quantify the benefit of using three instead of one camera? Similar how you quantified the importance of gray scale images?*

As explained in Section 4.4 and recalled in the conclusion, the main benefit of using 3 cameras is to get a more reliable classification as well as a confidence index which can be computed by merging the probabilistic information over the 3 views of the same particle. If the class identification is coherent between the 3 views, we increase the confidence index whereas if the classification is different on each view, we can flag the prediction as unreliable and discard this particle for further analysis. It is however difficult to quantify this benefit as the whole methodology was developed around an independent classification of each image provided by the instrument.

Concerning the importance of grayscale images, this importance is already quantified through a direct comparison of the classification scores with or without the use of the texture-based descriptors. This importance is relatively low for the hydrometeor type classification (Sec.4.1 : "It is interesting to see that among the 25 features kept, only 7 make use of the textural information given by the grayscale images. We attempted to simulate the results that we would obtain using a snowflake imaging device providing binary images of similar resolution by removing the texture-related features. By doing so, we obtained a HSS of 0.9 and a BER of 8.9%, hence indicating a very good performance using only geometric information."). It becomes however larger for melting snow detection (Sec.4.3 : "As for the riming degree, it seems that the textural features computed

from grayscale images have a significant importance for the classification, 3 of them being present in the 5 top-ranked descriptors. The HSS obtained after removal of the 8 textural descriptors is 0.76, 12% lower than initially.") and especially for riming degree estimation (Sec.4.2 : "When discarding the textural descriptors and running the classification again, we obtained a HSS of 0.49, 26% lower than initially. We can therefore conclude that the added value brought by the grayscale images is especially useful to detect and quantify the presence of riming.").

This quantification is given for information purposes only as the instrument pixel resolution is probably the parameter that affects the most the classification performance. Along this idea, a sentence has been added in the conclusion p.20 l.4-5 :

Future work in this direction will include the adaptation and evaluation of the model on airborne OAP and PIP images.

*Fig 2: Please define a.u. And can you draw a vertical line after 7 components in c)?*

A vertical line on the figure would decrease its readability. Alternatively, we added "$7^{th}$ component" below "6-fold symmetry" on the figure. We also modified the figure legend as follows :

(**c**) Fourier power spectrum corresponding to the angular signal above, in arbitrary units [a.u.].

*Fig 7: I would recommend to add titles to the subfigures: a) type classification, b) riming etc.*

We thank the reviewer for the suggestion. Figure 7 has been modified accordingly.

*Fig 8: x-axis is not in %.*

Figure 8 x-axis has been rephrased as :

Proportion of training data used

**Anonymous referee #3**

*Overall a well-conceived and well-written paper that uses machine learning to classify images from a Multi-Angle Snowflake Camera (MASC). Three sets of classification categories are used: 5 hydrometeor types, degree of riming, and whether a particle is melted or not.*

We thank the reviewer for their positive comment.

*A few areas need additional citations: 1) Page 2, Suggest adding reference to previous work on habit classification for optical probe data that used area ratio, the area of the particle divided by the area of the circle that circumscribes the maximum dimension of the particle. Relevant citations are Heymsfield and Parrish 1979; Heymsfield and Kajikawa 1987; Heymsfield and McFarquhar 1996; Heymsfield et al. 2002*

We added a sentence and additional references in the introduction on p.2 l.13 :

Other studies have classified optical probe data into eight habits based on an algorithm relating habit to particle maximum dimenstion and area ratio (e.g., Heymsfield and McFarquhar, 1996; Heymsfield et al., 2002).

*2) Page 4, There is another relevant Nurzynska et al. paper on shape parameters that needs to be cited, Meteorol. Appl. 20: 257265 (2013).*

We added the suggested reference in Section 2.3 p.4 l.27 :

Several of them have already been used for hydrometeor identification purposes in previous works (e.g., Lindqvist et al., 2012; Nurzyska et al., 2012, 2013; Grazioli et al., 2014; Schmitt and Heymsfield, 2014).

*Clarifications:*
*Given the limited depth of field of the MASC cameras, many particles in MASC images are out of focus. This has ramifications for any automated image processing algorithm.*

We thank the reviewer for raising this relevant point. The image processing and the outcome of the classification method is indeed influenced by the blurriness present in the images and this point was not commented in the first version of manuscript. In an effort to address this shortcoming, a few sentences have been added in Section 2.2, 3.2, and 3.6, as well as an appendix B. Please refer to the clarifications below for details.

*3) Please clarify how in focus (non-blurry) images were determined for both the training set and for the unlabeled MASC data. Do you use some objectively defined in-focus parameter? If so please define it and provide information on the distribution of this parameter for the training set and the unlabeled MASC data.*

In order to quantify the blurriness contained in MASC images, we introduced an empirical quality index $\xi$. See p.4 l.13-19 :

As the instrument triggering area is larger than the cameras depth of field, many MASC images are out of focus and appear blurred. This is important as the blurriness in the images will influence the value of certain descriptors introduced in Sec. 2.3 and in turn, the classification performance. Due to the large variety of shape and structure in the observed hydrometeors, it is however difficult to automatically distinguish out of focus images. For this study, efforts were made to establish a dimensionless empirical quality index $\xi$ based on particle size, brightness and interpixel variability which quantifies the blurriness present in the image. Typical values for $\xi$ lie within 7 (blurry) and 12 (sharp). The exact methodology used to develop this index as well as some illustrations are detailed in appendix B.

We also display in this appendix the distribution of the quality index for the training set and for unlabeled data.

*4) Please clarify what are the specific requirements for input images. Please include how sharp (in*

*focus) do the MASC particle images need to be for the method to be reliable.*

We try to include in the training set images of different qualities, from very bright and sharp snowflakes to slightly out of focus ones. However, we did not want to train the classification on blurrier images as it becomes difficult to assign a hydrometeor type/degree of riming to these particles, even by visual inspection. In practice, we found that this limit corresponds to a quality index $\xi \simeq 9$ (see Fig.15 in appendix B), so we included only images with $\xi \geq 9$ in the training set. Similarly, the classification method is reliable only on particles with a quality index above this threshold.

*5) Please clarify, is the intention and/or requirement that this method is applied to triplets of in-focus images.*

Following our response to question 4), images must have a a quality index $\xi \geq 9$ in order to be reliably classified. For triplets, images which are not satisfying this criterion are simply discarded and the classification is performed over the remaining pictures. This was formulated in Section 3.6 p.11 l.29-31 as follow:

As the three classification schemes have been trained on particle images described by a quality index $\xi \geq 9$, the predictions are expected to be reliable only for images above this threshold. For triplets, images which are not satisfying this criterion are discarded and the classification is merged over the remaining views.

*6) Regarding the training set (Section 3.2): What criteria were used to select images for the training set. Over how many storms and what range of environmental conditions.*

The criteria in terms of image blurriness have been explained in the response to question 4). The images included in the training set are coming from a broad range of storms selected randomly among the 2 available measurement campaigns. To clarify this point, we added the following sentences in Section 3.2 p.8 l.16-19:

In order to cover a large range of environmental conditions, hydrometeors included in the training set were selected from more than 40 snowfalls recorded in the Swiss Alps and in Antarctica. In terms of image quality, only particles characterized by a quality index $\xi \geq 9$ have been included in the training set. This threshold corresponds to a proportion of 50% of the whole dataset collected in Davos during the winter 2015-2016, as illustrated in appendix B.

Regarding blurriness issues and this quality index, a few clarifications have been added in Section 4.5 p.17 as well:

[...] every five minutes interval containing at least 30 non-blurry (i.e. $\xi \geq 9$) MASC images [...]
[...] to get rid of dark and/or blurry images, as described in Sec. 3.6.

*Section 4.5 Comparison with existing classification method. The differences between the instrument output of the MASC (33 um spatial resolution gray scale image) and 2DVD (0.2 mm spatial resolution binary image) as well as large difference in sample size for each 5 min interval (at least 30 non-blurry MASC images to 300 2DVD particles) makes direct comparison problematic. Very*

*close correspondence of classifications is not expected.*

The thresholds for the two instruments (at least 30 non-blurry images for the MASC and 300 particles for the 2DVD) have been selected in an effort to match the two sensors. Indeed, MASC sampling area is 4 times small than 2DVD and we show in Appendix B that around 50-70% of MASC images are filtered out by the quality index threshold. On the other hand, no filter is applied on 2DVD particles before performing the classification. If we take into account these two effects, 30 MASC images correspond roughly to 300 2DVD particles (30x4x2.5 = 300). We agree that these differences between MASC and 2DVD make the comparison problematic, but we still expect (and see) some consistency between the dominant type of hydrometeors observed by the two instruments. This is also a reason why we decided to perform the comparison over 5-minute intervals.

*7) The small particle classification into ice or rain relies on a temperature measurement. Should temperature be added to the characteristics in the machine learning.*

We agree with the reviewer, although one primary goal of the method is to have a classification output which is based exclusively on MASC (or another imaging instrument) output. If we add the temperature in the feature list, this is no longer the case. Moreover, it limits the method's scope of application to dataset which include this variable. For these reasons, we prefer to keep the classification as is, and potentially utilize *a posteriori* the information on particles fall speed and temperature (if available) to refine the classification output as, for instance, to further discern between liquid and solid small particles.

*8) Another possible reason for differences is that often ice particles of different classifications reach the surface simultaneously (Stark et al. 2013, MWR). It is possible that 30 MASC classified particles may not be representative of the 300 particles observed by the 2DVD.*

This is indeed a possible source of inconsistency which could explain part of the differences between HC-MASC and HC-2DVD. A sentence going in this sense has been added in Sec. 4.5 on p.17 l.24-26:

Stark et al. (2013) have observed that different ice particle habits could frequently reach the surface simultaneously. Given the large difference in the number of particles observed by both instrument, this phenomenon could also explain part of the discrepancy between the classification methods.

*9) Rather than showing only the one predominant classification for each 5 min period from 2DVD can the distribution of 1-min classifications be shown. This would provide a more direct comparison to distribution of classifications from MASC images.*

The suggested modifications are displayed on Fig.1 and Fig.2 below. The 2DVD classification over timesteps of 1min looks very similar to the one over timesteps of 5min (i.e. the one displayed in the manuscript). For this reason, we preferred to keep the version with 2DVD classification averaged over periods of 5min. In this manner, the presented time series are consistent between HC-MASC and HC-2DVD. Moreover, we think that this section is already quite complex (2 very similar but different classification schemes merged together, time aggregation to have a homogeneous time series for comparison purposes, etc.) and we are afraid that showing the HC-2DVD time series with a timestep of 1min (native) whereas both HC-MASC and the comparison are using timestep of 5min might confuse the reader.

*10) Page 18, for "temporal dynamics" do you mean "temporal variability"?*

Yes, we changed the word dynamics to variability in the manuscript (p.18 l.14).

*I am less concerned than other reviewer about not using fall speed as part of classification. When multiple particles are in one image, the one MASC measured fall speed cannot be clearly attributed to any one or set of those particles. Hence even if future versions of the algorithm use fall speed, it will not be available for individual particles cropped from MASC images with multiple particles.*

Yes, we agree that using fall speed would limit the classification to MASC images containing only a single particle on the frame. Aside the technical issues encountered with MASC fall speed measurement unit, this is one of the arguments that goes against the use this parameter in the classification algorithm.

[Figure]

Figure 1: Same as Figure 13 of the manuscript but with a time aggregation of 1 minute for HC-2DVD.

[Figure]

Figure 2: Same as Figure 14 of the manuscript but with a time aggregation of 1 minute for HC-2DVD.